# Structural and pKa Estimation of the Amphipathic HR1 in SARS-CoV-2: Insights from Constant pH MD, Linear vs. Nonlinear Normal Mode Analysis

**DOI:** 10.3390/ijms242216190

**Published:** 2023-11-10

**Authors:** Dayanara Lissette Yánez Arcos, Saravana Prakash Thirumuruganandham

**Affiliations:** Centro de Investigación de Ciencias Humanas y de la Educación (CICHE), Universidad Indoamérica, Ambato 180103, Ecuador; daya.yz.97@gmail.com

**Keywords:** SARS-CoV-2, HR1, principal component analysis, mutations, constant pH molecular dynamics, nonlinear and linear—Normal Mode Analysis

## Abstract

A comprehensive understanding of molecular interactions and functions is imperative for unraveling the intricacies of viral protein behavior and conformational dynamics during cellular entry. Focusing on the SARS-CoV-2 spike protein (SARS-CoV-2 sp), a Principal Component Analysis (PCA) on a subset comprising 131 A-chain structures in presence of various inhibitors was conducted. Our analyses unveiled a compelling correlation between PCA modes and Anisotropic Network Model (ANM) modes, underscoring the reliability and functional significance of low-frequency modes in adapting to diverse inhibitor binding scenarios. The role of HR1 in viral processing, both linear Normal Mode Analysis (NMA) and Nonlinear NMA were implemented. Linear NMA exhibited substantial inter-structure variability, as evident from a higher Root Mean Square Deviation (RMSD) range (7.30 Å), nonlinear NMA show stability throughout the simulations (RMSD 4.85 Å). Frequency analysis further emphasized that the energy requirements for conformational changes in nonlinear modes are notably lower compared to their linear counterparts. Using simulations of molecular dynamics at constant pH (cpH-MD), we successfully predicted the pKa order of the interconnected residues within the HR1 mutations at lower pH values, suggesting a transition to a post-fusion structure. The pKa determination study illustrates the profound effects of pH variations on protein structure. Key results include pKa values of 9.5179 for lys-921 in the D936H mutant, 9.50 for the D950N mutant, and a slightly higher value of 10.49 for the D936Y variant. To further understand the behavior and physicochemical characteristics of the protein in a biologically relevant setting, we also examine hydrophobic regions in the prefused states of the HR1 protein mutants D950N, D936Y, and D936H in our study. This analysis was conducted to ascertain the hydrophobic moment of the protein within a lipid environment, shedding light on its behavior and physicochemical properties in a biologically relevant context.

## 1. Introduction

Simulations in the realm of biological systems play a pivotal role in unraveling the mysteries of molecular movement and the analysis of critical variables. These simulations are essential for obtaining reliable approximations of physiological conditions, shedding light on intricate processes occurring within living organisms. One of the compelling areas where computational techniques prove indispensable is in understanding the structural dynamics associated with the transition from the pre-fusion to the post-fusion state of the SARS-CoV-2 spike protein (SARS-CoV-2 sp). This transformation unfolds within mere milliseconds [1], making it virtually impossible to capture experimentally. Computational methods, capable of simulating the precise physiological conditions and structural motions characterizing such protein transitions, emerge as a paramount tool in deciphering these rapid and elusive phenomena. To achieve this, it is crucial to grasp the intrinsic and physiological factors that impact protein structures, leading to conformational alterations and unfolding during the transition from the pre-fusion to post-fusion state of viral proteins. There is ample evidence that intracellular pH is critical in the viral infection mechanism. This influence is especially noticeable in the endosomal environment, which initially maintains a pH of 6.3 and then drops to below 6 during the SARS-CoV-2 viral cycle, meanwhile, late endosomes at pH of 5.5, emphasizing the importance of pH regulation in several stages of viral infection [2] as critical for protein unpacking [3]. The unfolding and fusing of the glycoprotein hemagglutinin with the host membrane depend on pH, according to earlier studies on influenza virus envelope proteins [4]. Moreover, low pH conformational alterations have been reported, indicating that strain-to-strain variations may exist in the pH threshold (5 to 7) at which these changes take place and, therefore, in the kinetic parameters [5]. In rhabdovirus, conformational changes in receptor-mediated membrane fusion were observed at an acidic pH below 6.4 [6]. Nevertheless, there is minimal in silico support for these experimental results of pH-dependent conformational changes of this protein, the infection rate at pH 6.2 is confirmed by in vivo research on the impact of pH on the infectivity of SARS-CoV-2 [7]. When simulating a low pH, it was found that the fusion loop of SARS-CoV-2 is held together by a disulfide bond [8] and undergoes a large conformational change at a low pH [9]. Our study aimed to use computational methods to interpret the effects of acidic physiological conditions on protein stability and the transition between the pre-fusion and post-fusion states of SARS-CoV-2 sp. This was done in light of previous findings that showed limited computational studies describing the effect of pH on protein conformation. It is worth mentioning that perturbation techniques have been used as a method for protein models using Normal Mode Analysis (NMA) [10,11] were used to visualize the internal motion of the open and closed states of SARS-CoV-2 sp [10,12,13] to find the globular motions [14] associated with conformational states [15] expressed by a globular minimum of potential energy [16]. Observation with Anisotropic Network Model (ANM) confirms that the Receptor Binding Motif (RBM) within the SARS-CoV-2 sp undergoes pronounced fluctuations, indicating an increased likelihood of interaction with the ACE2 receptor even when the protein is in its resting state [17].

Implementation of nonlinear analysis enhances the accuracy of the normal modes of a large molecular system, consequently improving the reliability of various stages within the protein transition models [18]. The structure that allows coronavirus to enter the host cell is the spike (S)-glycoprotein [19], a homotrimer that protrudes from the viral capsid and can trigger viral activity, and this consists of two functional subunits (Figure 1a) S1 and S2 [20]. S1 contains the N-terminal domain (NTD) [21] and the receptor binding domain (RBD) [19,22,23]. S2 holds the following subunits (i) fusion peptide (FP), (ii) heptad repeat 1 (HR1), (iii) central helix (CH), (iv) connector domain (CD), (v) heptad repeat 2 (HR2), (vi) transmembrane domain (TM), and (vii) cytoplasmic tail (CT) [21] and plays a central role in binding the spike to the host cell receptor [19,24], as it mediates the structural change that the spike protein undergoes during host cell interaction. Because of its importance for its conformational changes, our studies focus on understanding how and under what conditions these changes occur. In addition, the HR1 segment has a helical stalk that plays an important role in the S2 segment of SARS-CoV-2 [25], where a change in glycosylation sites can directly affect the infection rate of the virus and its ability to invade host cells [26]. Recent empirical data highlight the importance of glycosylation in HR1 and HR2 in regulating viral fusion and virulence [27], and how mutations at the N-glycosylation sites have structural effects on the integrity of the entire protein surface [28]. As a support to this observation, a study is also carried out on mutations and variations in the HR1 region aiming on understanding the structural dynamics of SARS-CoV-2 sp [29]. Numerous studies analyzing genome sequences have identified the structural effects of the new mutations in the HR1 region [30,31,32] (as shown in Figure 1a,b), which may affect the stability of the protein and allow flexible and dynamic unfolding and binding between virus and host membranes. Frequent mutations in the genomic sequence of the SARS-CoV-2 sp have resulted in new variants that are more prevalent than previously reported strains, such as delta (B.1.617.2) [33] and alpha (B.1.1.7) [34], which significant mutations that resulted in rapid territorial spread due to increased transmissibility. In particular, the mutations in the spike region in the delta variant, namely D950N, as shown in Table 1, are unique to this variant and were not present in previous variants such as alpha, beta, gamma, and omicron. The mutation in HR1, specifically D950N (Table 1 and Figure 1b,c) [35] has been shown to be a crucial mutation in the fusion process affecting the pathogenicity of the delta variant.

Non-synonymous mutations are thought to reduce protein stability. In particular, mutation D936H in the HR1 region (see Table 1) has been identified as causing a reduction in structural stability, with values ranging from −0.61 to −0.94 determined by docking and binding free energy (DDG) [36]. These destabilizing mutations may have an impact on the protein’s interaction with its receptor on the host cell.


Figure 1Organization of functional domains in SARS-CoV-2 (**a**), pre-fusion states (PDB: 6VYB) (**b**) and post-fusion states (PDB: 6XRA) (**c**): The N-terminal domain (NTD), receptor-binding domain (RBD), 685 (S1/S2) protease cleavage sites, fusion peptide (FP), heptad repeat (HR1), central helix (CH), connector domain (CD), heptad region 2 (HR2) transmembrane domain (TM), and cytoplasmatic tail (CT). The sequence of the HR1 of this study is shown in a grey box. The trimer chains of the SARS-CoV-2 sp are depicted in different colors: purple for chain A, green for chain B, and yellow for chain C. The structure of HR1 used for this study is highlighted in yellow for the pre-fusion state and in purple for the post-fusion state. 11 identified mutations are shown at the corresponding position and color coded in the HR1 structure.
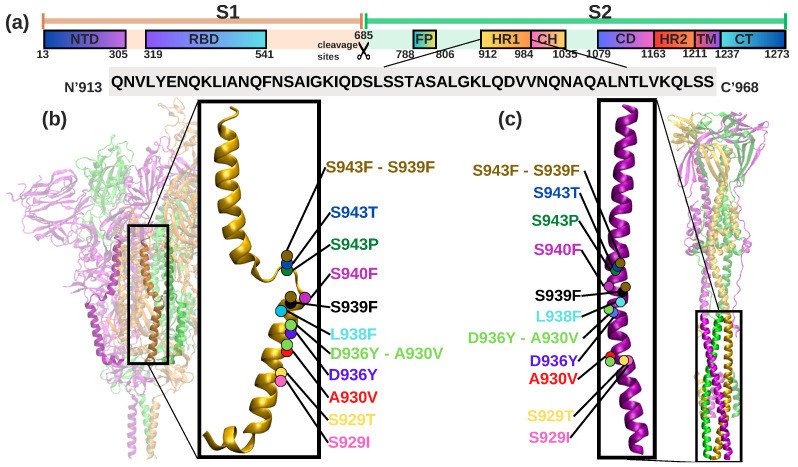



Our study primarily focuses on understanding the conformational changes of HR1 in the SARS-CoV-2 sp (Q913${}_{N_{terminal}}$ to S986${}_{C_{terminal}}$). Using (i) Principal Component Analysis (PCA) to analyze a subset of 131 structures from a collection of 250 spike proteins, all subject to various inhibitors, we study the conformational space and its relationship to intrinsic dynamics to explore the role in defining ligand binding pathways that can be used for inhibitor design. (ii) ANM was used to study the reliability and functional importance of low-frequency modes in adapting to different inhibitor binding (iii) Both linear and nonlinear NMA schemes were used to extrapolate the motions of HR1 from instantaneous linear and angular velocities, to understand the comparative structural fluctuations of eight different mutants of pre and post-fusion state of HR1. The aim is to analyze the dynamic role of HR1 of the spike protein within the viral process. In addition, (iv) cpH- MD simulations are implemented to provide an analysis of pKa values indicating that HR1 undergoes conformational changes toward its post-fusion structure at lower pH values. These analyses provide valuable insights into the dynamic behavior and structural variability of the HR1 structure. (v) Finally, we examine the hydrophobic regions in the pre-fusion state of the HR1 protein to determine its hydrophobic moment.

**Table 1 ijms-24-16190-t001:** Mutation reported for HR1 region of spike protein.

No.	Mutation	Description
1	D950N	Promotes membrane fusion to host membrane [37]
2	D936H	Generally present in Asia and Oceania [38]
3	D936Y	Most numerous and frequently occurring mutations [26]

## 2. Results and Discussion

### 2.1. PCA

PCA interprets the relationship between conformational space and intrinsic dynamics of the SARS-CoV-2 sp in its open state, particularly in relation to ligand binding pathways (Figure 2). Here to note that, focusing on the chain A subgroup, included residues 22 to 1127 to maintain conformational stability. For the analysis, a subset of 131 structures from a dataset of 250 structures from the Protein Data Bank (PDB) [39] was considered based on their structural similarity to the reference structure with an overlap of 80% or more. These structures were then projected onto the subspace defined by the primary axes PC1 and PC2, as shown in Figure 2. Subsequent examination revealed that a cluster of 57 structures (red) showed the presence of glycosidic linkages, a cluster of 34 structures (blue) contained inhibitory binding, and a cluster of 36 structures (light blue) matched the reported mutations. Figure 2 illustrates the representation of these structures in the PC1-PC2 subspace and highlights the different types of bonds they possess.

Hence, Figure 2 shows the 131 structures of SARS-CoV-2 sp. When comparing the pattern of components 1 and 2 of the PCs, groupings based mainly on their binding properties were observed among the analysed structures. These can be effectively divided into two distinct groups: those that exhibit glycosyl binding and those that exhibit inhibitory binding, which occurs when the structure is bound to an antibody. It is important to highlight that PC1 represents the primary direction of variance, followed by PC2. It is intriguing to observe how the dataset structures are distributed within the subspace defined by PC1 and PC2. This distribution enables us to distinguish or group the conformations according to their significant structural similarities or differences. This fundamental observation served as the basis for subsequent predictive modeling, structural heterogeneity, and identification of the optimal structures of SARS-CoV-2 sp. RNA dynamics and binding have shown that the first three eigenmodes of the corresponding eigenvectors exhibit remarkable dominant motions, with 52% of the total variation mainly accounted for by the wild type and 68% by the mutants. This different behavior could shed light on the structural rearrangements triggered by RNA binding [40]. Table 2 displays the frequency values associated with the lowest frequency calculations derived from ANM, commonly referred to as soft modes. To note, PCA structures and associated values are includesd in the Appendix A. Remarkably, our results showed that PC1 exhibited the greatest variability of the PC sets, contributing 85.11% of the total variance, while PC2 contributed 8.17% (see Table 2), with a total of 93.28% of the total covariance of the complex contained in the 2 first components of PCA. A similar trend was observed for the reported SARS-CoV-2 structures, and the top 2 PC components had a collective variance of 80% [41].

The structural variations captured by the first two PCs are visually represented in Figure 3. The color-coded eigenvector trajectories illustrate the conformational changes associated with these PCs (green eigenvector trajectories). The left panel shows the first PC, resulting from the analysis of the experimental structures with PCA. The order of PCs have been reordered to highlight their structural agreement with the ANM predictions (red eigenvector trajectories). In particular, PC1 shows a strong correlation (0.72) with ANM2 and exhibits similar structural behavior. Similarly, PC2 shows a high correlation (0.52) with ANM1, indicating a comparable pattern of structural changes.

We performed analyss of the 8 softest modes from the ANM, along with the corresponding PCAs, to explore the correlation between the modes predicted by the ANM, as well as, the experimental structures predicted by the PCA [42,43,44]. To assess the effectiveness of ANM, the modes were generated and their distribution was analyzed in relation to the corresponding PCs. In the case of PC1 and ANM2, the structures along these two axes have closely aligned (Figure 4a), indicating the equivalence of these modes, as indicated in Table 2. The distribution of the ensemble along the corresponding PCs was then examined by projecting it onto the ANM modes, the arrangement of structures along PC1-ANM2 is seen in Figure 4b, and the corresponding eigenvector projections in structure Figure 4a yielded a correlation coefficient of 0.72. This strong correlation underscores the reliability and functional importance of the low-frequency modes in binding structurally diverse inhibitors as well as mutant and glycosylated structures. These observations highlight the larger interface and lower frequencies associated with the projection of the ANM and PCA components and provide valuable insight into their functional role.

Having identified the primary modes that are strongly correlated with each other, the first 8 smoothest modes of PCA and ANM can be seen in the heat map (Figure 5b) for the open state of SARS-CoV-2. The strongest correlation (*r* = 0.72) is seen by the overlap of mode pairs between PC1 and ANM2. In addition, high correlation indices were observed between modes PC2 and ANM1. Similarly, modes PC8 and ANM7, and PC7 and ANM8 showed significant correlation. In addition, a high correlation was observed between modes PC2 and ANM1 (*r* = 0.52). Likewise, modes PC6 and ANM7 (*r* = 0.53), as well as PC6 and ANM8 (*r* = 0.48), showed a significant correlation. It is worth noting that pairs with correlation indices of less than 40% were considered irrelevant for this study. The specific correlation index shown in Table 2 and Figure 3.

In addition, a significant accumulation of glucoside-linked and a substantial number of inhibitor-linked structures were observed, suggesting that the structural changes undergone by the spike protein after initial recognition are driven by intrinsic behavior, independent of the presence or absence of inhibitory bound. In this study, these outcomes were employed to designate the open state structure for subsequent analysis, specifically the wild-type structure, aligning with the findings from the PCA and ANM. In contrast, Majumder et al. [17] studied the closed state structural fluctuation of SARS-CoV-2, SARS-CoV, and MERS-Cov with ANM showed high mobility of the residues of SARS-CoV-2. Since these experiments were conducted in the closed state, so it does not have higher mobility between its residues, especially the structures SARS-CoV and MERS-CoV remain immobile. To ensure that, a comparative study was performed to evaluate the mobility of the residues in the open and closed states of the SARS-CoV-2 sp. Figure 6 shows network structures indicating the mobility index, where blue represents the residues that exhibit mobility. PC1 mode, the open structure shows higher fluctuations (329.57 Å (Figure 6a) compared to the closed structure (297.73 Å (Figure 6b). Similarly, significant differences are observed between the open and closed structure in PC2 mode (18.35 Å (Figure 6a) and 12.46 Å (Figure 6b), respectively). The open structure in the ANM1 mode exhibits less variation (11.98 Å (Figure 6a) than the closed structure (15.08 Å (Figure 6b). In ANM2 mode, comparable variations are noted between the open and closed structures (5.47 Å (Figure 6a) and 5.63 Å (Figure 6b), respectively). These differences can be attributed to the compactness of the closed structure and the restricted movement between residues, which is essential for maintaining stability under physiological conditions. This suggests that the closed structure has lower mobility and higher stability in its closed conformation.

### 2.2. Conformational Flexibility and Structural Stability of HR1 Transition

Figure 7a shows the Root Mean Square Fluctuation (RMSF) of the structures of HR1 with 11 different mutations in the pre-fusion state. The L938F mutation in the fusion core of HR1 shows a significantly higher degree of residue flexibility compared to the other structures analyzed. In particular, this mutation showed a larger RMSF range, from a minimum of 2.57 Å (residue S941) to a maximum of 10.98 Å (residue Q913). Conversely, the S940F mutation had the lowest residue flexibility, as evidenced by oscillating RMSF values from 2.97 Å (residue S941) to 8.31 Å (Q913). Among the mutations studied, the S943P mutation had the highest peak with a value of 7.14 Å at residue G932. A significant range of shifts in residues A922 to Q957 is observed for all mutations examined. The residues with the greatest degree of flexibility, from Q913 to I923, are depicted in Figure 7b in relation to the post-fusion RMSF. In mutation S929I, a substantial shift is evident, with a range of 9.08 Å at residue N914 and a minimal shift of 0.9 Å at residue S939. Overall, the structures demonstrate limited variability in displacement, ranging from 5.30 Å at residue S968 (corresponding to mutation D936Y) to 0.60 Å at residue S940 (corresponding to mutation S940F). It is emphasized that the mutation exhibiting the lowest shift of its residues during the simulation is the S939F-S943F combination, with a range of 4.99 Å at residue Q965 and a minimum of 2.43 Å at residue S940.

The Root Mean Square Deviation (RMSD) (depicted in Figure 8a), which relates to the HR1 pre-fusion and the other associated mutations, confirmed a remarkable degree of similarity between the mutant structures. It can be observed that the studied structures converge to an equilibrium value of about 18 Å, with the highest RMSD value of 20.06 Å attributed to mutation S145T and the lowest value of 17.11 Å to mutation S943P. First 350 conformational steps of the simulation, observed in all the mutational structures studied, there is a noteworthy increase in the distance between steps, which remains stable from this conformation. The trajectory comprising 106 modes in which the 350 conformational transition frames indicates the increased RMSD, the similar pattern was also found in all mutants. In contrast, the RMSD values obtained from the post-fusion structure (Figure 8b) shows a progressive increase during the entire range of the simulation. In particular, the D936Y and S939F mutations show the largest RMSD of ≈20 Å, noted from conformations obtained from the transitional frames (0 to 1500). In contrast, the structure of the S939F-S943F double mutant shows notable stability with the shortest average shift in RMSD of ≈8 Å visible in transitional frames starting from 377 to 1500.

### 2.3. Linear vs. Nonlinear-NMA of HR1

Pre- and post-fusion of HR1, nonlinear and linear mode analyses were used to estimate conformational changes. The pre-fusion state of HR1 was used as the reference structure in the first study (Figure 9 linear (a) and non-linear (b)).

After performing a linear NMA for the pre-fusion state of HR1, we found an RMSD value of 7.30 Å. Subsequently, the reference structure was changed to the post-fusion state, resulting in an RMSD of 4.85 Å Figure 9a. This study allowed us to understand the momentum and flexibility of HR1 in its pre-fusion conformation. The structure predicted by the linear NMA exhibits a wider range of displacements, leading to instability under large simulation conditions. In contrast, the trajectory predicted by nonlinear NMA exhibits steady and stiff motion, resulting in better stability throughout the simulation. The Appendix A contain information on the NMA structures and frequencies used.


Figure 9Linear (**a**) and nonlinear (**b**) NMA for HR1. Initial (red), and final (blue) structure of transition. The intermediate states between the initial and final states are shown in gray.
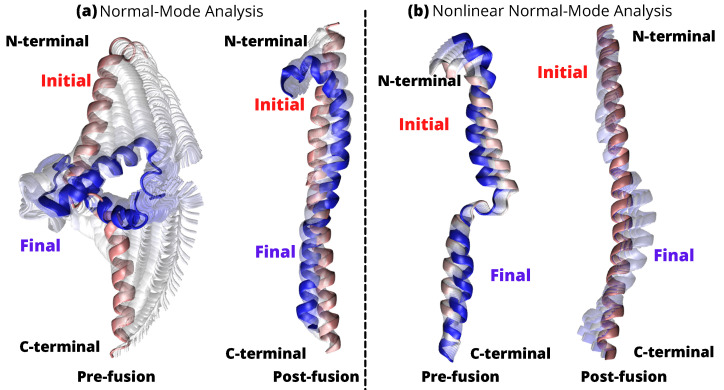



In the following analysis, we restored the reference structure to its pre-fusion state and examined the structural changes in detail. The calculated RMSD was 2.32 Å Figure 9b, indicating a small extent of deviation from the initial structure. Surprisingly, the nonlinear NMA approach resulted in a more stable and stiff structure, allowing a realistic evaluation of the interactions in a physiological context. Recently, the S proteins of SARS-CoV-1 and SARS-CoV-2 were studied by cryo-electron microscopy (cryo-EM). Ref. [45] remarkable conformational differences in the pre-fusion RBDs, confirming that the RBD of SARS-CoV-2 has a larger surface area and exhibits significant local conformational changes at specific amino acid residues. These structural differences help to enhance the interactions between the SARS-CoV-2 RBD and hACE2, which play a critical role in viral entry. These results indirectly support our observation of the (Figure 9a) intermediate transition states that are showing grey colored linear NMA, that reflect ts the existence of conformational changes.

Modes with higher frequencies require more energy to induce structural shifts than modes with lower frequencies. As a result, as the energy requirement increases, the likelihood of a shift decreases. In a broader sense, the system exhibits more pronounced shifts along lower frequency or slower modes, which correspond to gradual collective or extensive conformational changes. Higher frequency modes, on the other hand, primarily account for fast local motions. Appendix A depicts the structural changes observed for the first 10 lowest modes as defined by the linear and nonlinear NMA. We see that the first 15 frequencies calculated with the nonlinear NMA generally have lower values. For the structure from the pre-fusion state, the frequencies range from 0.000103042 cm${}^{-1}$ to 0.00226008 cm${}^{-1}$ (lowest and highest frequencies), while the post-fusion structure has a lower frequency of 0.000263012 cm${}^{-1}$ and a higher frequency of 0.0028324 cm${}^{-1}$ (Figure 10a). In contrast, linear NMA yields frequency values from greater than 0.176 cm${}^{-1}$ to less than 0.019 cm${}^{-1}$ (excluding the first 6 modes) for the structure in the pre-fusion state and from greater than 0.03 cm${}^{-1}$ to less than 0.23 cm${}^{-1}$ (excluding the first 6 modes) for the structure in the post-fusion state (see Figure 10b). In contrast, the linear NMA, the frequencies without the first 6 modes are distributed as follows, state before fusion: more than 0.176 cm${}^{-1}$ to less than 0.019 cm${}^{-1}$, state after fusion: more than 0.03 cm${}^{-1}$ to less than 0.23 cm${}^{-1}$ (see Figure 10b).

### 2.4. Prediction of pKa Values For HR1

The unprotonated fractions at various pH levels, as well as the ongoing estimates of pKa values, were evaluated to assess the convergence of protonation-state sampling and pKa values (Table 3). The titration curves presented in Figure 11 illustrate the pKa values derived from the corresponding titration curves. Our 22.5-ns cpH-MD of HR1 mutants; D936H, D936Y, and D950N, confirms the ability of single-pH to accurately predict the experimental apparent pKa values of interconnected sites in HR1. It shows that the simulations account for numerous protonatable sites in HR1 for each variant analyzed and that multiple sites contribute collectively to the macroscopic pKa values and titratable sites. It is important to note that as the simulation system size decreases, the potential of the bulk water phase shifts toward a negative value. This phenomenon artificially enhances the likelihood of accepting protonation attempts and diminishes the probability of deprotonation attempts within the neMD/MC algorithm. To ensure consistency with the probability ratio between protonated and unprotonated states as defined by the Henderson–Hasselbalch equation. Consequently, ionizable residues have higher probability of protonation, resulting in an apparent upshift in the pKa values.

The pKa values shown in Table 3 and outlined in Figure 11 exhibit a high degree of precision. They accurately reflect the reference values and have a small deviation that rarely exceeds 0.2 pH units. This result confirms the reliability and robustness of cpH-MD in predicting pKa values with precision. The Appendix A contain information on the specific pka values that were used. It is important to note that the simulations used in this study follow a conservative approach that allows for large error bars that are two and a half standard deviations from the mean, in practice, the pKa values estimated by CpH-MD simulations are generally accurate within a range of about 0.3 units.

Recently, a computational method to derive the electrostatic properties of the S proteins in SARS-CoV and SARS-CoV-2 [47] revealed that the RBDs of both proteins have positively charged interfaces that allow favorable interactions with the negatively charged surface of the hACE2 receptor. In account to that, our pH-dependent calculations of the relative folding energy for the RBDs of SARS-CoV and SARS-CoV-2 were observed to be most stable in pH range 6 to 9 indicating an optimal pH environment for their structural integrity. The pH dependence of binding energies showed that the complex structures formed by hACE2 and the S proteins of SARS-CoV/SARS-CoV-2 remain stable in a pH range of 7.5 to 10.5 [47]. These results highlight the ability of both variants to adapt to a similar pH environment and shed light on their survival strategies. Given the satisfactory agreement observed in our study, we decided to use the previously obtained model parameters to perform protein titration simulations.

Our findings suggest that the attachment of HR1 to the host cell membrane surface requires an acidic environment. This implies that in order to guarantee both efficacy and safety, upper pH safety limits for alkaline drugs and therapies are essential [48]. The pH was assessed using the three mutants’ titration curves. Importantly, the pH range of 4–6 is where the protonation of residues Glu-918 occurred (See Figure 11a and Appendix A). With a pka value of 4.5 in water, the structure is preserved at low pH, in particular the ionization of Glu had no effect on the shape of the protein. [49]. In addition, protonation of the Lysine-964 residues was observed at pH 8 to 10 (Figure 11b), with a corresponding pKa of 9.74 and remarkably for His-936 with a pka of 5.85 (Table 3) for the D936H. This result is consistent with studies under constant pH conditions, where neutral states of the catalytic dyads and histidine residues in the active site are essential for maximal enzyme activity, reflecting a pKa value of 6.9 for histidine-41 in the trimer of SARS-CoV-2 [50]. It is hypothesized that the greater deviation of pKa values in the all-atom constant pH simulations may be attributed to the reduced accuracy of the electrostatic interactions. The structural changes that occur at different pH values can be visualized in Appendix A. Lysine, being a basic amino acid, exhibits higher pKa values. For instance, in the case of Lysine-921, the pKa values are observed in the pH range of 9–10 for the D936H and D950N mutants, and between 9–11 for the D936Y mutant (Figure 11c). These values correlate with previous studies performed for lysine, in which lysine-164 plays a critical role in suppressing host gene expression, with an approximate pKa value of 9.74 [51]. Further, comparing the pKa values for residue Lys-933 in mutant D936H, mutant D950N, and variation D936Y, a similar pattern is also evident, with values of 9.50, 8.52, and 10.49, respectively, for Lys-921 (see Table 3). In addition, it should be noted that these computational studies may be useful in formulating the benefits and limitations of the alkaline extracellular environment and in establishing pH safety thresholds for the use of effective and efficient alkaline-based therapy.

### 2.5. Presence of an Amphipathic Helix in HR1 Wild Type (WT) and Mutants; D950N, D936Y and D936H

This study demonstrates the importance of amphipathic secondary helices in HR1 binding to membranes. Our findings (Figure 12) provide light on the critical aspects for understanding HR1-membrane contacts during fusion, which result in structural changes in the spike protein and fusion with the lipid membrane, influencing spike protein function. The presence of amphipathic phases within HR1 suggests their potential involvement in mediating interactions with other molecules or membranes, given their unique ability to simultaneously associate with hydrophobic and hydrophilic surfaces. Since the amphipathic helical conformation is vital for binding to the host cell membrane. An example is the conserved amino-terminal amphipathic alpha-helix, which is necessary for targeting regulators of G protein signaling proteins to the plasma membrane [52]. Mutations D950N, D936Y, and D936H were introduced to study the hydrophobicity <H>, leading to notable differentiation. D950N, D936Y and D936H mutants, collectively resulted in an overall charge of +1 for all these mutant analogs. This charge disparity is in stark contrast to the neutral charge (0) observed in the HR1 (WT) peptide. As a consequence of these mutation-induced changes, the structural characteristics of the HR1 were noticeably different. This was evident in the hydrophobicity <H> values, which provide a quantitative measure of amphipathicity. Specifically, for the sequence section 16-NSAIGKIQDSLSSTASAL-33 (WT), the D936Y and D936H mutants exhibited <H> values of 0.397 and 0.351, respectively. These values represented a significant increase in amphipathicity compared to the wild-type HR1 peptide, which had a <H> value of 0.301, as illustrated in Figure 12. Similarly, the sequence 34-GKLQDVVNQNAQALNTLV-51 (WT) that introduced changes in the sequence section for the D950N mutant, led to a slightly higher <H> value of 0.311, as compared to the wild-type <H> value of 0.301. These alterations underscore the substantial impact of these mutations on the structural features of HR1 peptides, particularly in terms of their amphipathic character as indicated by the hydrophobic moment.

Conversely, it is worth noting that the introduced mutations in the HR1 peptides may lead to the emergence of distinct structural features. In particular, these alterations were observed to have an impact on the hydrophobic moment <μH>, manifesting as a reduction in the values for the D936Y and D936H mutants when compared to the HR1 (WT) peptide. Specifically, for the sequence section 16-NSAIGKIQDSLSSTASAL-33 (WT), the <μH> for the D936Y mutant measured at 0.366, while for the D936H mutant, it was 0.412. In stark contrast, the wild-type HR1 peptide exhibited a higher <μH> of 0.461, as visually represented in Figure 12. Similarly, in the case of the D950N mutant, the introduced changes were found to yield slightly lower <μH> values when compared to the wild-type HR1 peptide. Specifically, for the sequence section 34-GKLQDVVNQNAQALNTLV-51 (WT), the <μH> for the D950N mutant was observed to be 0.443, which was marginally lower than the <μH> value of 0.453 observed in the wild-type sequence. These findings suggest that the helix has a pronounced segregation between its hydrophobic and hydrophilic regions along its length, resulting in a distinct perpendicular arrangement of these two contrasting faces. These alterations emphasize the significance of the mutations in influencing the structural characteristics of the HR1 peptides and, notably, their interactions with biological membranes. In addition, this helix wheel diagram drawn for the single point mutation of HR1 helps to understand the variation in structural features such as RMSD and RMSF, as shown in Figure 8 and Figure 7, and their influence on the normal modes (Figure 9). Peptides with hydrophobic regions can disrupt lipid membranes and possibly form pores or destroy lipid bilayers [53,54]. Studies confirm that peptides with positive hydrophobicity can interact with viral hydrophobic surfaces [55]. Previous research suggests that the hydrophilic portion of the amphipathic helix plays a critical role in antiviral function and that any disruption of this element may reduce efficacy against influenza A virus (IAV) [56]. Prior studies has established the relevance of the hydrophilic location in the amphipathic helices of viral proteins, emphasizing our findings and the necessity to further discover and analyze these features.

## 3. Materials and Methods

### 3.1. Retrieve Dataset

A dataset of experimental structures was created to identify the inherent dynamics of the SARS-CoV-2 sp in the open state. To facilitate this analysis, a wild-type structure of the SARS-CoV-2 sp in the open state (PDB:6VYB [19]) was taken as a reference. In the same way, a wild-type structure of the SARS-CoV-2 sp in the close state (PDB:6VXX [19]) was taken as a reference to establish a comparison between open and closed states. Structural data obtained by electron microscopy and available in the Protein Data Bank (RCSB-PDB) [39] were compared to the reference structure. This was done to compile experimental structures demonstrating various functionalities, such as those in ligand-bound states with antibodies or nanostructures. The structures selected for this study are listed in Table 4. We selected structures with sequence alignment greater than 90% and structure overlap of 80% or more (see Table 4) compared to the respective reference structure. The experimental structures collected in the database were analyzed using the following procedure:

The mean position ($\Delta R_{i}^{s}=\left[\Delta x_{i}^{s}\Delta y_{i}^{s}\Delta z_{i}^{s}\right]T$) of each structure of the dataset (*s*) was calculated with respect to the distance of each alpha carbon (1≤i≤N) to the alpha carbon of the of the reference structure (where each component $\Delta x_{i}^{s}=x_{i}^{s}-\langle x_{i}\rangle$) [57]. These steps were iteratively applied to each structure within the 250 structures of the initial dataset, enforcing a threshold of RMSD ≤0.001 Å. Structures not meeting this criterion were excluded, resulting in a refined dataset comprising 131 structures. For this purpose, we use the ProDy library (software version 2.0) [58] to generate the ensemble of PDB structures projected onto the library for further analysis.

### 3.2. Preparation of Pre-Fusion State Mutants

The HR1 mutations selected for this study were introduced using CHARM-GUI [59] (www.charmm-gui.org, accessed on 17 January 2022) [59,60], a web-based platform specifically designed for the construction of complicated systems. “PDB Reader” [42] (www.charmm-gui.org/?doc=input/pdbreader, accessed on 17 January 2022) [42,59]. was used to model missing residues of the HR1 protein. To introduce mutations into the specific residues, we used the CHARM-GUI platform by selecting the “Mutation” option [42]. The following eleven mutations were considered for this study: 1. A930V [60], 2. D936Y [60,61], 3. L938F [62], 4. S929I, 5. S929T [26], 6. S939F [26], 7. S943F, 8. S940F [62], 9. S943P [63], 10. S943T [64,65], and a combination of two mutations commonly found together, 11. S943F-S939F, shown in Figure 1, in addition to the mutations listed in Table 1. Our assessment is centered on the 11 identified mutations of HR1, with a specific emphasis on comparing their conformational states between the pre and post fusion forms.

### 3.3. PCA of the Ensemble

The main modes in our analysis were derived by decomposing the covariance matrix C for the dataset based on the open and closed states of the spike protein SARS-CoV-2 sp. The covariance matrix (Cov(σ,p)) is a 3N*3N matrix that represents the relationships between dimensions and variable pairs. When considering the covariance matrix (Cov(σ,σ)=Var(σ,p)), its own variance is significant, and the primary aim was to diagonalize it to obtain individual variable variances. Given the commutativity of covariance (Cov(σ,p) = Cov(p,σ)), the covariance matrix demonstrates symmetry along the diagonal. This decomposition was calculated using the equation $C=\sum_{i=1}^{3N}\sigma_{i}p^{\left(i\right)}p^{(i)T}$, where $p^{\left(i\right)}$ represents the eigenvector corresponding to the ith eigenvalue $\sigma_{i}$. The eigenvalues and eigenvectors were obtained by this decomposition process, where $\sigma_{i}$ represents the eigenvalue associated with the largest variance component. To enhance the alignment of the assembly, the optimization process involves the following steps: (i) A random superposition was conducted utilizing the Kabsch algorithm. (ii) From the preceding step, an ensemble of mean coordinates is derived, referred to as the “average coordinate”. (iii) Subsequently, the Kabsch algorithm was repeated, utilizing a pair of superimposed structures generated from the “average coordinates”. (iv) Steps (ii) and (iii) were iterated until a mean model was attained, with an RMSD distance constraint between them of 0.001 Å. The main modes, labeled PC1, PC2, etc., are determined by a ranking process. PC1 corresponds to the direction of maximum variance in the dataset, followed by PC2 and subsequent modes. These main modes provide valuable information about the structural variability within the ensemble of structures. In particular, we were interested in visualizing the distribution of structures in the dataset in the subspace defined by PC1 and PC2. In this way, we can effectively distinguish or cluster the conformations based on their distinct structural similarities or dissimilarities. This alignment determines the mean positions $<Ri>={\left[<x_{i}><y_{i}><z_{i}>\right]}^{T}$ for the$\;C_{\alpha }$. These mean positions represent the average coordinates of the $\;C_{\alpha }$ in the entire ensemble of structures and provide valuable insight into the structural properties of the system. The ensemble files, denoted by the Normal Mode Wizard extension (NMWiz) (software version 1.2) [58], contain coordinate data related to the normal modes and the reference frame. To visualize these NMWiz files [58], the Visual Molecular Dynamics (VMD) (software version 1.9.4a55) [66,67] was used to project the vector components of PCA and the ANM, as well as the interaction network [57,68]. For the correlation analysis, the highest-ranking correlations of the PCA and ANM modes were determined. Then, the resulting structures were projected onto the PC1 and PC2 modes, to facilitate visualization and to create plots, the Matplotlib (software version 3.1.1) tool [69] was adopted.

### 3.4. ANM Analysis and Overlap with Modes of PCA

ANM was applied to the open spike protein of SARS-CoV-2 sp and compared with PCA previously performed on experimental structures. In ANM, the Hessian matrix H was decomposed into a set of 3N−6 eigenvalues $\left(\lambda_{1}\right)$ and the direction of the corresponding eigenvectors $u_{1}$. The ANM covariance $C_{ANM}=H^{-1}$ such that, $1/\lambda_{1}$ serves as the counterpart to PCA $\sigma_{1}$, and $u^{\left(i\right)}$ from the ANM serves as the counterpart to $p^{\left(i\right)}$ of PCA. The overlap of the modes from ANM and PCA modes were given by the cosine correlation $O_{ij}=p^{\left(i\right)}.u^{\left(j\right)}$ [57]. The internal vectors are described by the ANM vectors, and the superposition ensures that the translation and rotation of the rigid body structures remain unchanged. The matrix *C* was diagonalized to determine the main modes involved in the changes of the structures observed during the experiments ($p_{pca}^{\left(i\right)}$) [57]. The stand-alone ProDy package (software version 2.0) [58] was used to calculate ANM and provided theoretical B-factors for each residue. These B-factor values were averaged within protein segments, such as secondary structure, to obtain flexibility values for these segments. To predict coordinated movements within the protein, ANM values were used to map the cross-correlation between residues.

### 3.5. Linear Normal Mode Analysis (Linear-NMA)

The NMA for the HR1 form of the SARS-CoV-2 sp was performed using NOMAD-Ref [70] (lorentz.immstr.pasteur.fr/nomad-ref.php, accessed on 17 January 2022) [71]. In this method, the matrix of the second-order derivative of the potential energy, V, was calculated at a local minimum. The vectors of the normal modes were determined by solving the eigenvalue problem of the system $A^{T}VA=\lambda$. The normal modes, which represent the motion patterns, were determined by the eigenvectors ($A_{k}$) and their corresponding eigenvalues ($\lambda_{k}$). The eigenvalues provide us with useful information about the energy required to cause displacements along the direction of the corresponding eigenvectors. Using the NMA, it is possible to determine the energy required for different protein conformations. Therefore, in subsequent analyzes, provided preference to the modes with the lowest energy requirements for the protein conformation. NOMAD -Ref was used as a method to compare ANM developed by Kim et al. [70] to estimate the distances between residues. In our analysis, we chose an elastic constant of 5 Å and an average RMSD of 1 Å for the calculated trajectories. In addition, ANM cutoff of 1 Å for a total of 106 modes were considered.

### 3.6. Nonlinear Normal Mode Analysis (NonLinear NMA)

The pre- and post-fusion transitions of HR1 were calculated using the nonLinear-NMA implemented in NOLB software (software version 1.9) [18]. It involves a nonlinear extrapolation of the instantaneous motion directions described by the normal modes obtained in the subspace of block rotations and translations, to sustain the angular velocity of the HR1, it can be interpreted as the outcome of an implicit force, implying that the movement of the residues can be regarded as a rotational motion around a specific center. Modes were determined by diagonalizing the mass-weighted stiffness matrix projected onto the subspace of rotations, translations, and blocks, and the computation involves the diagonalization of the rotations-translations of blocks(RTB)-projected mass-weighted stiffness matrix $K_{w}=\left(P\tilde{L}\right)\tilde{\Lambda }{\left(P\tilde{L}\right)}^{T}$, Where $\tilde{L}$ is the matrix that possesses the RTB normal modes together with the corresponding diagonal eigenvalue matrix $\tilde{\Lambda }$. To note, diagonalization of $K_{w}$ projected by RTB results in a set of eigenvectors that represent the instantaneous linear ($\vec{v}=M_{b}^{1/2}\vec{v_{w}}$) and angular velocities ($\vec{w}=I^{-1/2}{\vec{w}}_{w}$) of individual rigid blocks.

### 3.7. cpH-MD Protocol

A hybrid nonequilibrium molecular dynamics/Monte Carlo (neMD/MC) constant-pH MD method [72], as implemented in NAMD software version 2.14 [73], was used, to determine the protonation states of titratable sites within a protein, by predicting the most likely protonated or non-protonated states. All simulations were performed using the CHARMM36 force field [74], and titratable amino acids were selected based on these parameters, covering a pH range of 2 to 14 with intervals of 1 unit, while keeping the parameters consistent. The simulation protocol involves in (i) preparing the initial structure, in which selected protein models were solvated in a water box with a 20 Å buffer between the protein surface and box boundaries. To neutralize the excess charge, Na${}^{+}$ and Cl${}^{-}$ ions were added to the system, along with Periodic boundary conditions, by employing particle mesh Ewald electrostatics and smooth switching of the Lennard-Jones forces with a cutoff of 10 Å was applied. Further, for the (ii) simulation phase, the solvated protein system was subjected to an initial energy minimization of conjugate gradient 2000 steps, followed by a 10 ns equilibration run under NpT ensemble conditions at a temperature (300 K) and a pressure of 1 atm. Final structures obtained in the equilibration phase served as the initial structure for the constant-pH MD (cpH-MD) production run, during the production phase, a time step of Δt= 2 ps was used, and temperature control (300 K) was achieved using a Langevin thermostat with a damping coefficient of 1/ps. Protonation state changes were attempted every 15 ps during the 22.5 ns (500 neMD/MC cycles), resulting a cumulative simulation time of 315 ns for each protein system. The pKa reference values were assigned using the Propka3 (software version 3.4.0) [46] and maintained throughout the simulations to ensure efficient sampling, although they did not affect the final simulation outcomes.

### 3.8. Amphipathic Helix Analysis

Heliquest prediction algorithm [75] (https://heliquest.ipmc.cnrs.fr/cgi-bin/ComputParams.py, accessed on 17 January 2022) was adopted to elucidate the structural configurations and helical properties of the pre-fusion state of HR1, particularly the hydrophobic moment <μH>, for helices found in both HR1, wild type and three mutant variants; D950N, D936Y, and D936H. The principal objective was to study the structural configurations and helical properties of the pre-fusion state of HR1, specifically focusing on characterizing amphipathic helices and assessing their physicochemical parameters, such as the <μH> and hydrophobicity <H>. These parameters are paramount in characterizing the amphipathicity of helical segments when adopting an alpha-helix conformation. <μH>, in particular, serves as a quantitative measure of amphipathicity by calculating the vector sum of hydrophobic side chains within the helical region. This comprehensive approach allowed us to gain deeper insights into the structural aspects of the HR1 pre-fusion state, enabling a thorough analyses of its helical features and alterations induced by the aforementioned mutations. This analysis serves as a benchmark or analogy to the two NMA mentioned above, where the amino acid in the amphipathic region of HR1 changes its globular motion for the first 15 modes.

## 4. Conclusions

We performed a PCA of 131 A-chain structures of the spike protein in the presence of various inhibitors. Carried out the conformational space and inherent dynamics of the protein, with a focus on understanding ligand binding pathways for inhibitor design. A correlation factor of 0.72 between PCA modes and ANM modes, indicates the reliability and functional importance of low-frequency modes in adapting to different inhibitor binding. To characterize the function of HR1 dynamics in viral processing, linear NMA and nonlinear NMA approaches are used. The linear NMA exhibited larger RMSD ranges (7.30 Å) and showed considerable variability between structures, whereas the Nonlinear NMA showed stability throughout the simulation (RMSD 4.85 Å). Analyses of the frequencies of the linear and nonlinear modes guarantee that the energy required for conformational changes in the nonlinear modes is much lower than the energy required for conformational changes in the estimated linear modes. Specifically, the frequency values for the linear modes showed a value of 0.000103042 cm${}^{-1}$ for their most stable mode, whereas the nonlinear modes showed a value of 0.019 cm${}^{-1}$ for their most stable mode in HR1 structures in their pre-fusion form. This corresponds to a difference of 0.018896 cm${}^{-1}$ in frequencies, indicating the stability of the medians of the nonlinear modes and lower energetic costs associated with their conformational changes. The single-PH simulations effectively predicted the pKa order of linked residues in HR1 mutations. Our cpH-MD simulations revealed conformational changes in HR1 at lower pH values, indicating a shift toward post-fusion structure. Consequently, the titration curves of HR1 mutants; D936H, D936Y, and D950N indicates numerous protonatable site, and the transition from fully protonated to fully deprotonated persists over a broad pH range (>5 for GLU 918 and >8 for LYS936/933/947/921). The pKa values determined in this study show the effects of low pH on protein structure. For example, a pKa value of 9.52 was determined for the Lys-921 residue in the D936H mutant, a value of 9.50 for the D950N mutant, and a slightly higher value of 10.49 for the D936Y variant. These results are in agreement with similar trends described in other studies. The introduction of mutations in the HR1 peptides has produced striking modifications in their structural attributes, particularly evident in the significant increase in amphipathicity compared to the wild-type HR1 peptide, characterized by a <H> value of 0.301. Notably, the D950N mutant displayed a slightly higher hydrophobic moment value of 0.213, underlining the impact of specific mutations on this crucial structural feature. These findings elucidate the consequential alterations in amphipathic properties brought about by mutations in the HR1 peptides, providing valuable insights into their potential implications for various biological processes and further highlighting the importance of hydrophobic moment values in the study of peptide functionality.

## Figures and Tables

**Figure 2 ijms-24-16190-f002:**
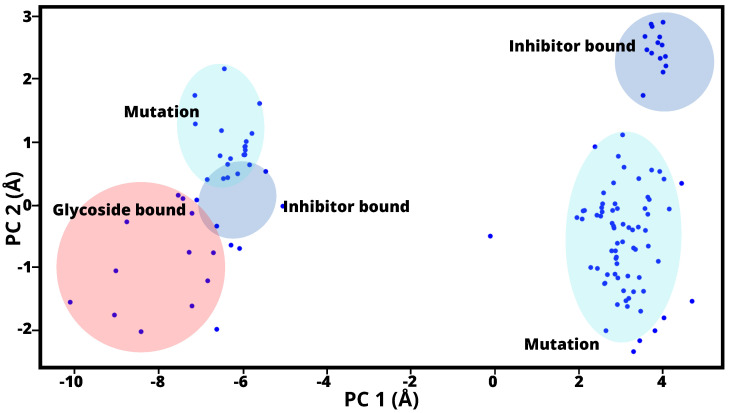
Projection of two-dimensional that illustrates the distribution of various SARS-CoV-2 sp structures based on the lowest-frequency modes of PC1 and PC2. Red Circles: These represent 57 structures with glycoside bound. Blue Points: There are 34 with inhibitor bound. Light-Blue Points: Surrounding the 36 data points in light blue are mutants of SARS-CoV-2 sp.

**Figure 3 ijms-24-16190-f003:**
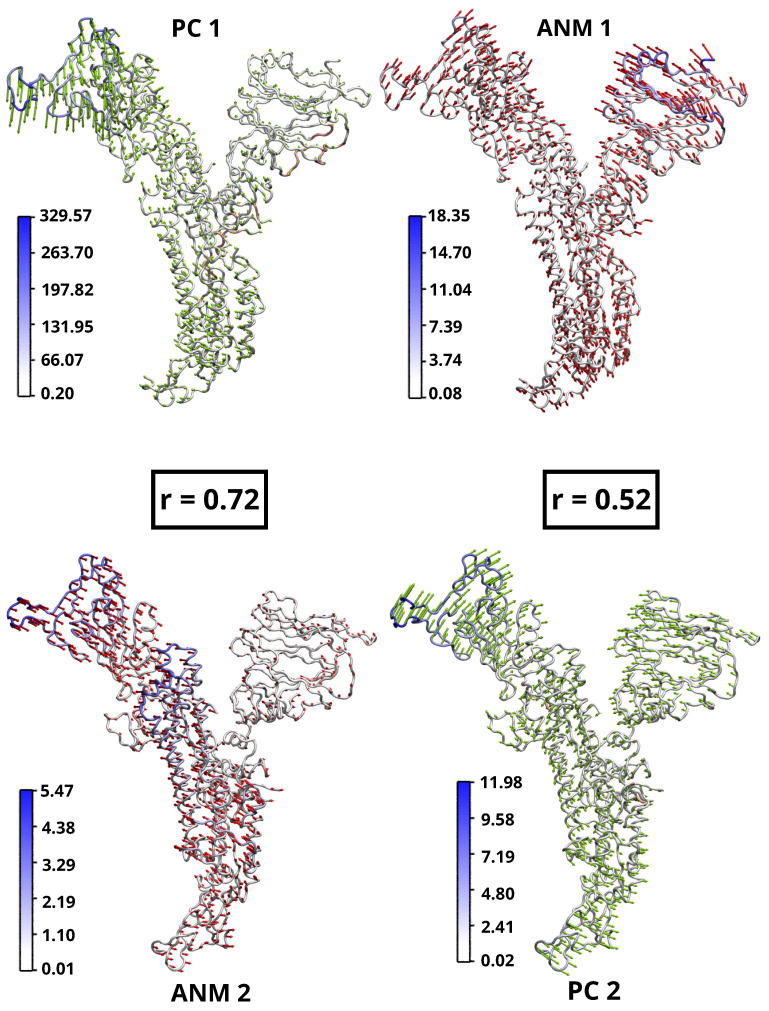
Comparison of PCA (PC1, PC2) and ANM (ANM1, ANM2) trajectories with eigenvector mapping to show the difference; white to blue colored bars represent the magnitude of residual motion in Å. Arrows indicate directions and lengths of eigenvectors corresponding to ANM (red) and PC (green).

**Figure 4 ijms-24-16190-f004:**
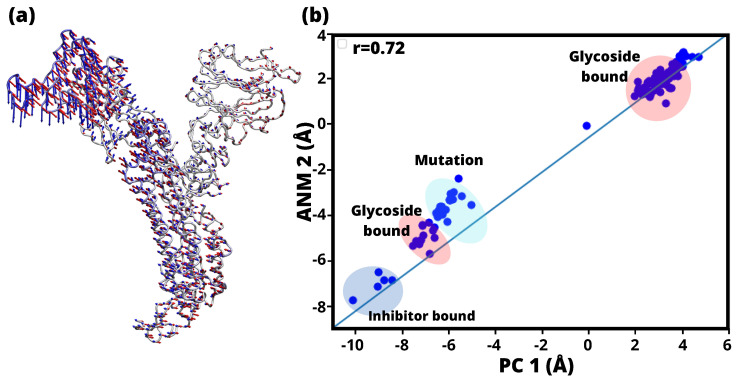
(**a**) SARS-CoV-2 sp monomeric conformation in the open state. Blue arrows: Eigenvector associated with PC1 capturing the dominant conformational change in the SARS-CoV-2 sp monomer. Red arrow: eigenvector based on ANM2. (**b**) Projection of 131 structures onto PC1 and ANM2: glycosidic linkages, inhibitor linkages, and mutants. Red circles: Glycosidic bond. Blue circles: Inhibitor bond. Light blue circles: mutants.

**Figure 5 ijms-24-16190-f005:**
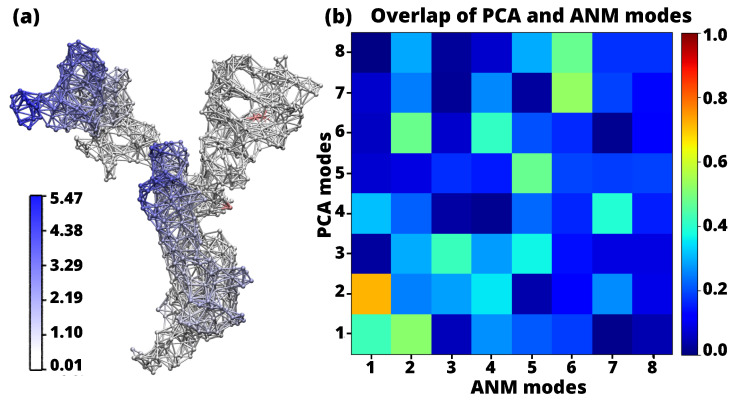
(**a**) The network model for the monomeric open state of SARS-CoV-2 of PC1 with corresponding values of the mobility indicated in the color bar by the square fluctuation of the residues are shown as blue and white zones in the network structure in a scale of 0.01 Å to 5.47 Å. (**b**) Overlap between the 8 PCA modes with the highest rank and the 8 ANM modes with the lowest rank. The orange square in the visualization indicates a strong correlation (0.72) between these modes.

**Figure 6 ijms-24-16190-f006:**
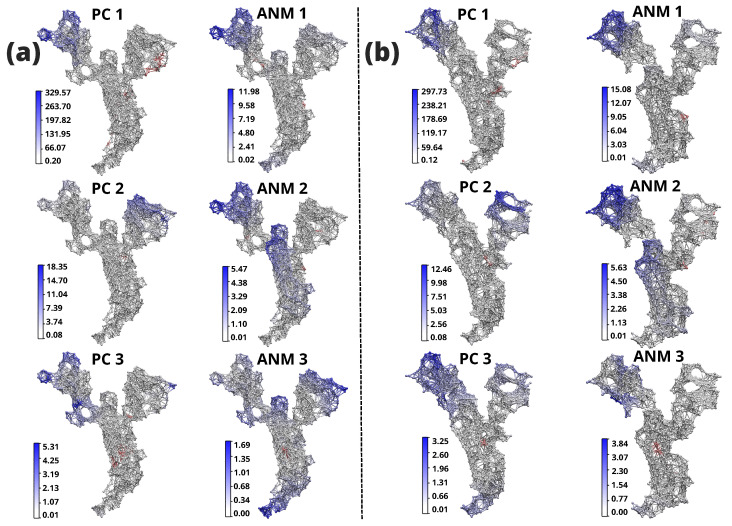
ANM representation of the SARS-CoV-2 sp monomer in the open and closed states. (**a**) SARS-CoV-2 monomer in the open state. On the left: PCA applied to the experimental structures. On the right: ANM representation of the theoretical structures. (**b**) SARS-CoV-2 monomer in the closed state. On the left: PCA applied to the experimental structures. On the right: ANM for the theoretical structures. For all generated structures, the residues showing mobility are highlighted in blue. The bar scale (white to blue) indicates the extent of their mobility in Å.

**Figure 7 ijms-24-16190-f007:**
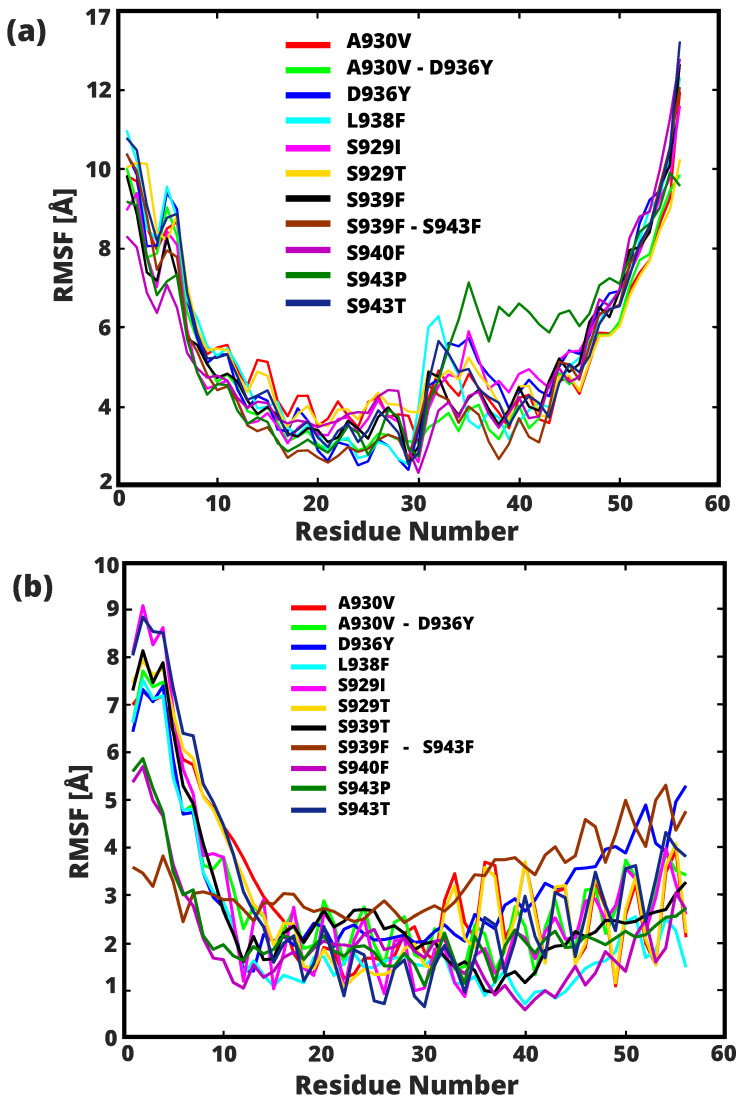
RMSF (Å) of C${}_{\alpha }$, showing flexibility in 60 residues of 11 mutants, from the pre- (**a**) and post- (**b**) fusion states. The y-axis represents the RMSF, while the x-axis corresponds to the amino acid residue numbers. All 11 analyzed mutations are characterized by a specific symbol coding scheme that allows the evaluation of fluctuations in the carbon alpha of the backbone during the simulation. The 11 mutants are marked with different colors.

**Figure 8 ijms-24-16190-f008:**
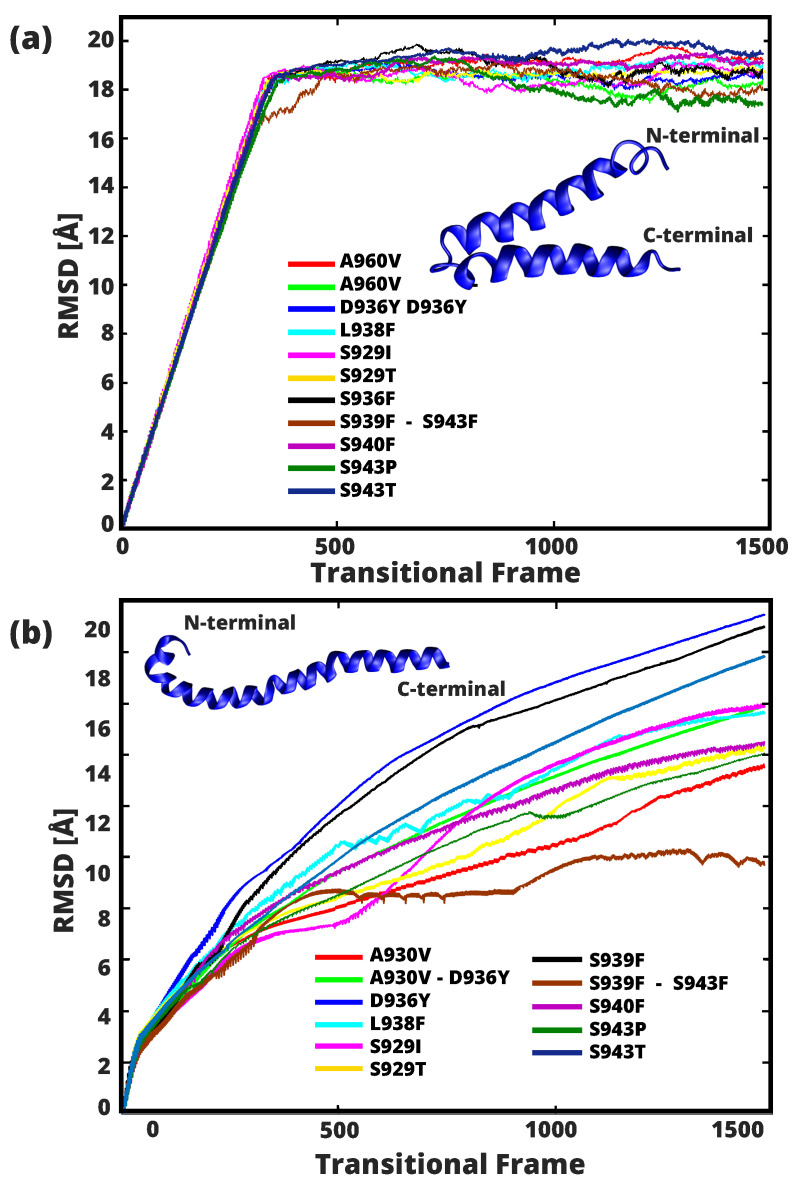
RMSD of backbone C${}_{\alpha }$ shows different flexibility levels between the 60 residues of HR1 and associated mutants, for the pre (**a**) and post (**b**) fusional states. All 11 analyzed mutations are characterized by a specific symbol coding scheme.

**Figure 10 ijms-24-16190-f010:**
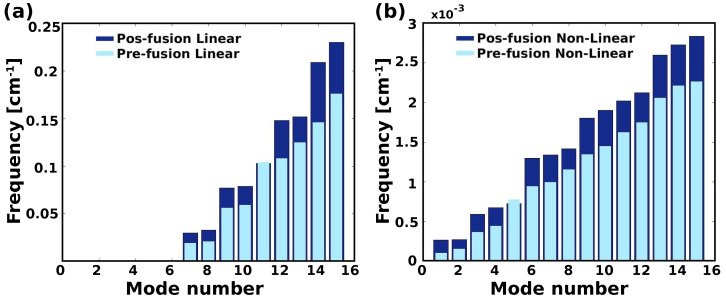
Distribution of mode number vs. frequency of the 15 lowest modes using (**a**) Linear-NMA and (**b**) NonLinear-NMA for the pre (light blue bars) and post (blue bars) fusion of HR1.

**Figure 11 ijms-24-16190-f011:**
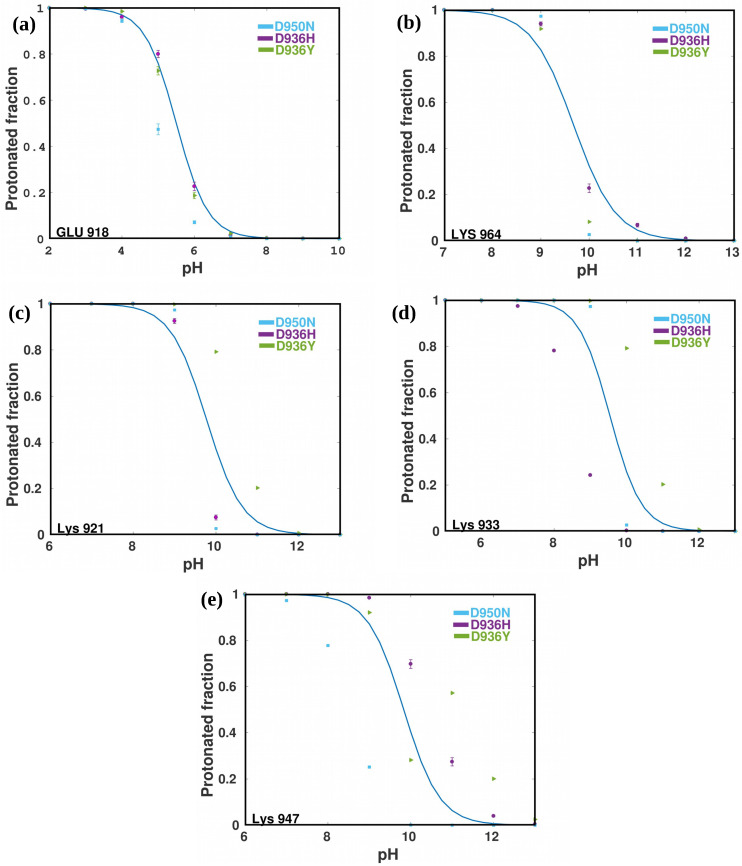
Protonated fraction vs pH of 5 residues of the three mutants (**a**) Glu 918, (**b**) Lys 964, (**c**) Lys 921, (**d**) Lys 933, (**e**) Lys 947, titrated in water. The dots represent the protonation ratio, the macroscopic titration curves for the mutant: D936Y (green), D936H (purple), and D950N (light blue). A fitting procedure (blue line) is based on the Henderson-Hasselbalch equation using pKa${}_{estimate}$ reference values (Table 3). The error bars indicate the presence of estimated 95% confidence intervals.

**Figure 12 ijms-24-16190-f012:**
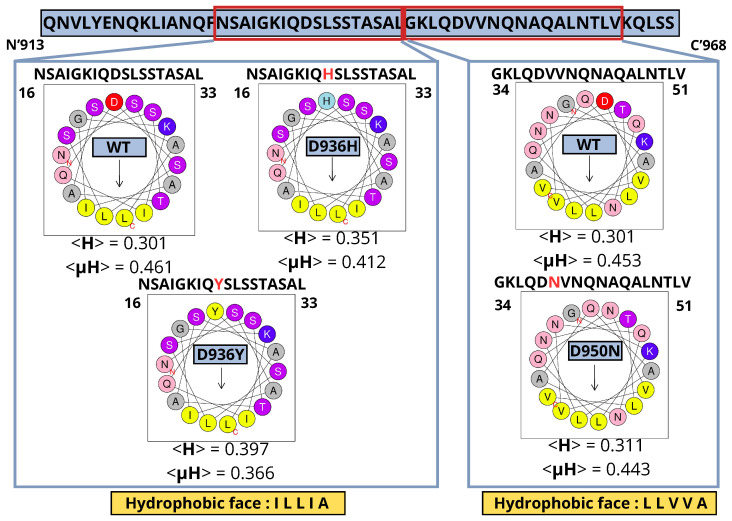
Sequence and helical wheel representations of amphipathic structures of HR1 and its mutants, The hydrophobic moment <μH> is denoted to quantify the amphipathicity of the helices, <H> represents hydrophobicity. Hydrophobic and aromatic residues (yellow), charged residues (blue), uncharged polar residues (indigo), and glycine (gray). The hydrophobic phase of the peptide is indicated by the arrows.

**Table 2 ijms-24-16190-t002:** Correlation overlap of the 8 slowest modes of PCA and ANM in the open state SARS-CoV-2 sp.

	ANM1	ANM2	ANM3	ANM4	ANM5	ANM6	ANM7	ANM8
	($6.26\times 10^{-4}$)	($1.11\times 10^{-3}$)	(**2.56** $\times \;10^{-3}$)	(**3.15** $\times \;10^{-3}$)	(**6.56** $\times \;10^{-3}$)	(**1.27** $\times \;10^{-2}$)	(**2.06** $\times \;10^{-2}$)	(**2.83** $\times \;10^{-2}$)
PC1 (85.11)	0.42	0.72	0.03	0.31	0.07	0.06	0.07	0.01
PC2 (8.17)	0.52	0.25	0.30	0.22	0.09	0.48	0.25	0.29
PC3 (2.11)	0.05	0.28	0.42	0.03	0.17	0.07	0.02	0.02
PC4 (1.68)	0.27	0.35	0.28	0.02	0.15	0.41	0.26	0.06
PC5 (0.54)	0.21	0.04	0.37	0.23	0.47	0.20	0.03	0.30
PC6 (0.52)	0.19	0.11	0.14	0.16	0.19	0.17	0.53	0.48
PC7 (0.34)	0.01	0.27	0.09	0.39	0.19	0.02	0.19	0.18
PC8 (0.23)	0.04	0.09	0.09	0.15	0.19	0.12	0.13	0.17

**Table 3 ijms-24-16190-t003:** The pKa values for the HR1 obtained from titration simulations. pKa${}_{estimate}$ values, which were determined using Propka3 (software version 3.4.0), and pKa${}_{vacumm}$ and pKa${}_{solvent}$ obtained from newMD/MC simulations [46].

Mutation	Group	pka${}_{\mathrm{estimate}}$	pka${}_{\mathrm{vacumm}}$	pka${}_{\mathrm{solvent}}$
D950N	ASP 936	5.26	3.8604 ± 0.0511	4.4001 ± 0.0523
	GLU 918	5.90	4.8754 ± 0.0565	4.9693 ± 0.0430
	LYS 921	9.77	11.1691 ± 0.0752	9.5000 ± 0.0553
	LYS 933	9.54	12.1578 ± 0.1182	9.5000 ± 0.0553
	LYS 947	9.83	10.3714 ± 0.0079	8.5296 ± 0.0237
	LYS 964	9.68	10.4851 ± 0.0529	9.5000 ± 0.0553
D936H	ASP 950	5.50	3.2198 ± 0.0511	4.5541 ± 0.0380
	HIS 936	6.702	5.1570 ± 0.0016	5.8596 ± 0.0629
	GLU 918	5.90	3.8604 ± 0.0298	5.4325 ± 0.0746
	LYS 921	9.77	10.3372 ± 0.0170	9.5179 ± 0.0213
	LYS 933	9.53	9.9543 ± 0.0023	8.5244 ± 0.0228
	LYS 947	9.82	11.4573 ± 0.0209	10.5839 ± 0.0767
	LYS 964	9.68	10.4612 ± 0.0755	9.7427 ± 0.1314
D936Y	ASP 950	5.51	3.6691 ± 0.0866	4.0672 ± 0.0809
	GLU 918	5.90	5.7565 ± 0.0755	5.6504 ± 0.1109
	LYS 921	9.77	10.8356 ± 0.0671	10.4949 ± 0.0192
	LYS 933	9.53	10.1694 ± 0.0919	10.4949 ± 0.0192
	LYS 947	9.82	10.9075 ± 0.0345	10.4167 ± 0.5285
	LYS 964	9.68	11.1651 ± 0.0880	9.5009 ± 0.0007

**Table 4 ijms-24-16190-t004:** Protein dataset of pre-fusion state spike and its mutants in the presence of Inhibitor bound, Glucoside bound and structures that preset any mutations on its sequence.

Inhibitor bound	6XCM	7CWS	7DZX	7L02	7ONA	7JWB	7BYR	7EJ4
	7DCC	7A25	6Z43	7AKD	7K85	7CWL	7C2L	7DL1
	7N9T	7CAI	7KMK	7E8C	7LRT	7MKL	7CHH	7KL9
	7L3N	7R8M	7SC1	7CAC	6NB6	7OAN	7FAE	7NS6
	7LD1	7P40	7K8S	7E3K	7E5R	7VNC		
Glucoside bound	6VYB	6VXX	6X79	7CN9	6XLU	6X6P	7BNN	6XF5
	7KDG	6VSD	6ZB4	7KDJ	6ZOW	7JJI	7E7B	6XR8
	7E7D	6ZP0	6ZP1	7KDK	7MTE	7KD1	7A4N	6ZWV
	7DX1	7KRQ	6ZOY	6ZOX	6XS6	7LWW	6XKL	7MJ9
	7LWI	7TLC	7N1U	7CAB	7LYK	7M8K	7N1Q	7EDF
	7K9H	7CN4	7CN8	6ZGE	5X58	6CRW	7SOB	6X2A
	7TLA	7KJ2	7LAA	7LQV	6ACC	7BBH	6ZGF	7SO9
	7SBP							
Mutation	7V8C	7SBK	7TOU	7SBS	7VX1	7Q6E	7V76	7LWS
	7V7N	7OD3	7SXW	7SXV	7V78	7FEM	7SXU	7V7D
	7N8H	7SXS	7SXT	7W92	7MJG	7SXR	7VX9	7KDI
	7FCD	7EAZ	7BNM	6ZP2				
	7DZW	6ZGG	6X29	7T9J	7QO7	7WK2	7TB4	7WK4

## Data Availability

Data are contained within the article.

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
