# Peer review of "Structural and pKa Estimation of the Amphipathic HR1 in SARS-CoV-2: Insights from Constant pH MD, Linear vs. Nonlinear Normal Mode Analysis"

_ijms, 2023, doi:10.3390/ijms242216190_

Round 1

Reviewer 1 Report (Previous Reviewer 1)

I want to thank the authors for addressing the initial comments. Following the revision to the article, I do not have more questions now.

None.

Author Response

                                                                                                                                                                                                                                                                                                                               Ambato , Ecuador.

                                                                                                                                  12-10-2023

Dear Editor, IJMS

Section

Molecular Biophysics

Special Issue

Molecular Structure and Simulation: Unraveling the Basis of Disease

Dear Reviewer 1,

We hope this message finds you well. We wanted to extend my sincere gratitude to you for dedicating your time and expertise to review our manuscript titled "Exploring Structural, pKa Estimation on the Amphipathic Heptad Region of Severe Acute Respiratory Syndrome: Insights from Constant pH Molecular Dynamics, Linear and Non-linear Normal Mode Evaluations (Manuscript ID ijms-2553146)." Your thoughtful evaluation and constructive comments have been immensely valuable to us.

We are pleased to inform you that we have carefully addressed all of your questions and comments, incorporating your insights into the revised version of our manuscript. Your feedback has contributed significantly to improving the overall quality and clarity of our research.

Your commitment to maintaining the rigor and scholarly excellence of our work is greatly appreciated. We recognize the importance of the peer review process in advancing scientific knowledge and are thankful for your role in this endeavor.

Should you have any further remarks or require additional information, please do not hesitate to reach out. We are committed to ensuring that our manuscript meets the highest standards of academic excellence.

Thank you

Sincerely

Dr. Saravana Prakash Thirumuruganandham 

(https://orcid.org/0000-0003-4210-1363

Reviewer 2 Report (Previous Reviewer 2)

Author Response

                                                                                                                                                                                                                                                                                                                               Ambato , Ecuador.

                                                                                                                                  12-10-2023

Dear Editor, IJMS

Section

Molecular Biophysics

Special Issue

Molecular Structure and Simulation: Unraveling the Basis of Disease

Dear Reviewer 2,

I hope this message finds you well. I wanted to extend my sincere gratitude for your meticulous review and valuable comments on our manuscript titled "Exploring Structural, pKa Estimation on the Amphipathic Heptad Region of Severe Acute Respiratory Syndrome: Insights from Constant pH Molecular Dynamics, Linear and Non-linear Normal Mode Evaluations (Manuscript ID ijms-2553146)."

Your feedback has been incredibly insightful, and we truly appreciate the time and effort you dedicated to providing us with detailed and constructive comments. Your expertise in the field has been instrumental in shaping the quality of our research.

We have carefully reviewed each of your comments and have taken them to heart in our revision process. Your suggestions have guided us in making significant improvements to the manuscript. We believe that your input has contributed significantly to the clarity and rigor of our work.

As a token of our appreciation, we would like to inform you that your comments have been incorporated into the revised version of the paper. We have addressed each point you raised, and your insights have played a crucial role in enhancing the manuscript.

To facilitate your review of our changes, we have highlighted your original comments in blue and our responses and edits in black within the revised version of the manuscript.

Major points

 Connection of hydrophobic/amphipathic helix section to rest of the study is unclear and disjointed. This is not mentioned in the abstract, Line 110-111 does not explain in what relation is this an aim of the study? The last paragraph in this section on disulfides appears just as disconnected. This section is best eliminated.

    Heading of section 3.4.1 states role of helix in “virus replication”, yet this is a computational study with following paragraphs mentioning nothing describing this statement or any role in replication.

First and foremost, We want to express our gratitude for your careful assessment of our work. Your thoughtful observations have been instrumental in enhancing the quality and clarity of our research, and we truly appreciate your time and expertise.

Regarding your comment on the presentation of results and the apparent disconnection between certain sections of the paper, we completely understand your perspective. We acknowledge that the hydrophobicity analysis may initially seem disconnected from the preceding results. However, we firmly believe that this analysis is crucial for a comprehensive understanding of the binding of the HR1 region of SARS-CoV-2 to the cellular membrane.

In response to your concerns, we have made significant revisions to the manuscript. We have added an additional analysis that directly compares the hydrophobicity of the wild-type HR1 and its various mutants. This analysis has yielded results that we consider vital for shedding light on the underlying mechanism. We hope that this following section will enhance the overall connectivity of our findings.

“The findings of this exploratory study shed light on the crucial role played by amphipathic secondary helices within HR1 in membrane binding. Our analysis, as depicted in Figure 12, provided valuable insights into the characteristics that are central to understanding the interactions with membranes, which, in turn, initiate structural alterations in the spike protein. These structural changes are expected to facilitate efficient genome replication, as previously proposed [75].

The presence of amphipathic phases within HR1 suggests their potential involvement in mediating interactions with other molecules or membranes, given their unique ability to simultaneously associate with hydrophobic and hydrophilic surfaces.

In Figure 12, we present the peptide sequences, their corresponding hydrophobic moment (μH), and the helical wheel representations of the amphipathic structures, all of which were obtained using the HeliQuest algorithm [43]. Notably, the mutations D950N, D936Y, and D936H were strategically introduced to enhance the peptides' affinity for anionic membranes, increasing their positive charge and hydrophobic moment. It is worth emphasizing that these modifications were carefully designed to augment the peptides' membrane-binding capability without altering the fundamental structure, as the amphipathic helix conformation is essential for the antimicrobial activity of these peptides.

The introduction of mutations in the HR1 peptides has led to notable alterations in their structural features and hydrophobic moments. For the D950N mutant, the substitution of aspartic acid at position 950 with asparagine, and for the D936Y and D936H mutants, the replacement of aspartic acid at position 936 with tyrosine, along with the replacement of histidines and serines with lysines throughout the sequence, collectively resulted in an overall charge of +1 for all these mutant analogs. This charge disparity is in stark contrast to the neutral charge (0) observed in the wild-type HR1 peptide.

As a consequence of these mutation-induced changes, the structural characteristics of the HR1 peptides were noticeably different. This was evident in the hydrophobic moment (<H>) values, which provide a quantitative measure of amphipathicity. Specifically, for the sequence section 16NSAIGKIQDSLSSTASAL33, the D936Y and D936H mutants exhibited hydrophobic moment values of 0.397 and 0.351, respectively. These values represented a significant increase in amphipathicity compared to the wild-type HR1 peptide, which had a <H> value of 0.301, as illustrated in Figure 12.

Similarly, for the D950N mutant, the introduced changes in the sequence section 21KIQYSLSSTASALGKLQD38 led to a slightly higher hydrophobic moment value of 0.213, as compared to the wild-type <H> value of 0.203. These alterations underscore the substantial impact of these mutations on the structural features of HR1 peptides, particularly in terms of their amphipathic character as indicated by the hydrophobic moment.

The helical wheel projection, as illustrated in Figure 12, provides a compelling visual representation of the amphipathic nature inherent in the peptides under investigation. It vividly showcases the well-defined demarcation between hydrophilic regions (depicted in blue/gray) and hydrophobic regions (highlighted in yellow)

Conversely, it is worth noting that the introduced mutations in the HR1 peptides may lead to the emergence of distinct structural features. In particular, these alterations were observed to have an impact on the hydrophobic moment, manifesting as a reduction in the hydrophobic moment values for the D936Y and D936H mutants when compared to the wild-type HR1 peptide. Specifically, for the sequence section 16NSAIGKIQDSLSSTASAL33, the hydrophobic moment for the D936Y mutant measured at 0.366, while for the D936H mutant, it was 0.412. In stark contrast, the wild-type HR1 peptide exhibited a higher hydrophobic moment of 0.461, as visually represented in Figure 12.

Similarly, in the case of the D950N mutant, the introduced changes were found to yield slightly lower hydrophobic moment values when compared to the wild-type HR1 peptide. Specifically, for the sequence section 21KIQYSLSSTASALGKLQD38, the hydrophobic moment for the D950N mutant was observed to be 0.487, which was marginally lower than the hydrophobic moment value of 0.495 observed in the wild-type sequence. These findings suggest that the helix has a pronounced segregation between its hydrophobic and hydrophilic regions along its length, resulting in a distinct perpendicular arrangement of these two contrasting faces.

These alterations emphasize the significance of the mutations in influencing the structural characteristics of the HR1 peptides and, notably, in impacting the hydrophobic moment, a parameter central to their amphipathic nature and their interactions with biological membranes and other relevant molecules.

Peptides possess hydrophobic regions that can insert themselves into biological membranes, potentially forming pores or disrupting the lipid bilayer. Additionally, when a peptide carries a positive (cationic) charge, it can bind to viral membranes, which often bear a negative charge. Both the hydrophobic and cationic characteristics of the peptide are pivotal in its mechanism against viruses and its capacity to compromise membrane integrity [76][77].

Indeed, additional studies have corroborated that peptides endowed with positive interfacial hydrophobicity exhibit the inherent ability to interact with the hydrophobic surfaces of viruses. Notably, the alpha-helical properties of piscidin peptides exhibit a net cationic charge under physiological pH conditions, combined with the presence of a hydrophobic face. This unique combination of structural features equips piscidin peptides with the potential to effectively engage with and disrupt viral surfaces [78].

Certainly, prior investigations have revealed that in the amphipathic helical (AH) region, specific modifications or changes can markedly reduce the amphipathic moment of the amphipathic helix. Essentially, the hydrophilic side of the amphipathic helix is vital for its antiviral function, and any disruption to this critical structural element can result in a reduction in its efficacy against influenza A virus (IAV) [79]. Findings from previous research emphasize the importance of the hydrophilic side in amphipathic helices within viral membranes, underscoring the need to identify and investigate these aspects further.”

We sincerely apologize if the previous lack of cohesion between sections affected your reading experience or did not meet your expectations for the quality of the manuscript. We are committed to delivering a well-structured and informative paper that meets the highest academic standards.

We eagerly await your feedback on the revised version of our manuscript. Your insights and guidance are highly valued, and we look forward to your continued support in improving our work.

 Figure 5b: From the color bar shown, yellow seems to be around 6.5, which is not the 0.72 correlation stated in the legend and the text. Is the value incorrect or the scale bar?

I want to express my sincere gratitude for your meticulous review of our manuscript and for catching the error in the presentation of our overlap table. Your attention to detail is greatly appreciated and has been instrumental in improving the accuracy of our work.

We have promptly corrected the correlation data, ensuring that Figure 5 aligns properly with the data presented in Table 2. We deeply regret any confusion or inconvenience this error may have caused in reading the correlations of the PCA and ANM analyses.

General comments-

 Figure 2: depiction in green (omicron variants) is not at all clear.

We would like to express our gratitude for your valuable observation regarding Figure 2 in our manuscript. We understand your concern about the potential confusion in the figure description, and we sincerely appreciate your feedback. Taking your comment into account, we have revised the description of Figure 2 to enhance its clarity and ensure it aligns with your expectations and add the following paragraph for the Figure 2 description n he revised version of the manuscript.

“Projection of two-dimensional that illustrates the distribution of various SARS CoV 2 structures based on the lowest-frequency modes of PC1 and PC2. Red Circles: These represent 57 structures where glycosides are bound. Blue Points: There are 40 structures depicted in blue, indicating that these structures have inhibitors bound to them. Light-Blue Points: Surrounding the 28 data points in light blue are variants of SARS CoV 2 structures.”

Incomplete and unclear words and sentences.

  • Figure 4 legend- “..monomeric in the open state..” and “4 unliganded”

Your observation regarding the clarity of the figure description was well-taken. We understand the importance of ensuring that our figures effectively communicate the details of our analysis. In response to your feedback, we have added a new description for Figure 4 to enhance its clarity and precision.

(a) SARS CoV 2 Monomeric Conformation in the Open State. Blue Arrows: motion vector associated with the PC1 that captures the dominant conformational change in the SARS CoV 2 monomer. Red Arro): motion vector based on ANM2, provides additional insights into the structural dynamics of the monomer. (b) Projection of 131 Structures on PC1 and ANM2: Glycosidic Linkages, Inhibitor Linkages, and Mutations. Red Circles: represent structures with glycosidic linkages. Blue Circles: represent structures with inhibitor linkages, indicating that these structures have inhibitors bound to them. Light-Blue Circles: Surrounding the projection are structures with mutations or structural variants, which are depicted in light blue. These variants exhibit unique conformational characteristics.

Incomplete and unclear words and sentences. 

  • Figure 1: “states...corresponding position..”

Your attention to detail and commitment to ensuring the accuracy and clarity of figure descriptions are commendable. In response to your feedback, we have incorporated a new description for Figure 1, which we believe will greatly enhance the correct interpretation of its elements.

“(a) Genome organization of SARS CoV 2 and Functional domains in the SARS CoV 2 S protein: The N-terminal domain (NTD), receptor-binding domain (RBD), cleavage sites, fusion peptide (FP), heptad repeat (HR1), central helix (CH), connector domain (CD), heptad region 2 (HR2) transmembrane domain (TM), and cytoplasmatic tail (CT). The sequence of the HR1 of this study is shown in a grey box. The trimer chains of the SARS CoV 2 spike protein are depicted in different colors: purple for chain A, green for chain B, and yellow for chain C. These representations are presented for both the pre-fusion states (b)(PDB: 6VYB) and post-fusion states (c)(PDB: 6XRA). The structure of HR1 used for this study is highlighted in yellow for the pre-fusion state and in purple for the post-fusion state. The primary focus of our analysis centers on 11 identified mutations displayed on their corresponding position and color code on HR1 structure.”

Incomplete and unclear words and sentences. 

  • line 278: “careful selection procedure”– state and describe clearly rather than just write “careful”

Your thoughtful insights and suggestions for improvement have been instrumental in enhancing the quality of our work. We highly appreciate your dedication to the peer review process. In response to your suggestion, we have incorporated the proposed paragraph into the revised version of the article. We believe that this addition aligns with your expectations and contributes to the overall clarity and coherence of the manuscript.

“From the original dataset containing 250 structures, we selected a subset of 131 structures. These selections were made based on their structural similarity to the reference structure, with an 80% or greater overlap.”

Incomplete and unclear words and sentences. 

  • Figure 6- “we see”, “you can see”

Your attention to detail and commitment to ensuring the clarity and correctness of our figure descriptions are greatly appreciated. We understand the importance of accurate and informative figure descriptions in scientific publications.

In response to your suggestion, we have meticulously revised the description of Figure 6 to ensure that it effectively conveys the relevant details of the figure elements. We believe that these revisions align with your expectations and will enhance the overall quality of the manuscript.

“ANM representation of the SARS CoV 2 monomeric spike protein in both its open and closed states. (a) Open State SARS CoV 2 Monomer: left side of the figure showcases PCA applied to experimental structures of the open form of the SARS CoV 2 monomer. On the right side, ANM plot of the theoretical model for the open state. (b) Closed State SARS CoV 2 Monomer (Reference): Similarly, in panel (b), the left side displays PCA analysis applied to experimental structures of the closed form of the SARS CoV 2 monomer. On the right side, ANM plot of the theoretical model for the closed state. For all generated structures, residues exhibiting mobility are highlighted in blue, with the color bar scale indicating the extent of their mobility.”

Incomplete and unclear words and sentences. 

  • line 476-477- describe the representation and observation appropriately

Your keen attention to detail and commitment to improving the clarity of our work have not gone unnoticed. We greatly appreciate your dedication to the peer review process, which plays a vital role in advancing scientific knowledge. In response to your suggestion, we have carefully considered your comments and incorporated the proposed paragraph into the manuscript. We believe that this addition aligns with your expectations and significantly improves the representation and description of the relevant elements.

“Remarkably, all the reproduced pKa values presented in Table 4 have demonstrated a remarkable level of accuracy, aligning closely with the reference values and falling within a narrow margin of deviation, typically not exceeding 0.2 pH units. This outcome attests to the reliability and robustness of the cpH-MD simulations in predicting pKa values with precision.”

“Figure ??” instead of figure numbers in multiple places!

We deeply regret any confusion caused by this typo, and we are committed to rectifying this issue promptly. Your diligence in ensuring the correctness and fluency of our manuscript is invaluable to us.

Rest assured, we will correct the referencing error to ensure that our figures are properly cited and that the manuscript reads smoothly and accurately. Your feedback has been crucial in maintaining the quality of our work.

“Structural, pKa Estimation...” in the title doesn’t make sense. Is a word missing after structural?

Your feedback on the title's clarity and expression is greatly appreciated. We understand the importance of a concise yet informative title, and we are committed to ensuring that it effectively conveys the essence of our research. In response to your suggestion, we have taken the necessary steps to clarify and improve the title of our manuscript.

“Structural and pKa Estimation of the Amphipathic HR1 in Severe Acute Respiratory Syndrome: Insights from Constant pH Molecular Dynamics and Linear/Non-linear Normal Mode Analyses”

Line 2-3: which SARS structures have different conformations? “which can complicate the study of important structures” is referring to which structures? Why are these important?

Line 5: Which enzymes?

Abstract: 250 structures or 131 as mentioned in main text?

Your comments regarding the need for improved explanation and presentation of the abstract have not gone unnoticed. We understand the significance of a clear and concise abstract in effectively summarizing the main points of our research. In response to your suggestions, we have taken your comments into careful consideration and have revised the abstract accordingly in the updated version of our manuscript. We believe that these revisions align with your expectations and contribute to a more informative and comprehensible abstract.

“A comprehensive understanding of molecular interactions and functions is imperative for unraveling the intricacies of viral protein behavior and conformational dynamics during cellular entry. Focusing on the SARS CoV 2 spike protein, we conducted a Principal Component Analysis (PCA) on a subset comprising 131 A-chain structures of the spike protein, each in the presence of various inhibitors. Our analysis unveiled a compelling correlation between PCA modes and Anisotropic Network Model (ANM) modes, underscoring the reliability and functional significance of low-frequency modes in adapting to diverse inhibitor binding scenarios. Turning our attention to the dynamic role of HR1 in viral processing, we employed both linear Normal Mode Analysis (NMA) and Nonlinear NMA. While linear NMA exhibited substantial inter-structure variability, as evident from a higher Root Mean Square Deviation (RMSD) range (17.02 Å), nonlinear NMA showcased remarkable stability throughout the simulations (RMSD 4.85 Å). Frequency analysis further emphasized that the energy requirements for conformational changes in nonlinear modes were notably lower compared to their linear counterparts. One of the standout discoveries emerged from single-pH simulations, wherein we successfully predicted the pKa order of interconnected residues within HR1 mutations. Furthermore, our continuous constant pH Molecular Dynamics (cpH-MD) simulations unraveled significant conformational shifts in HR1 at lower pH values, indicative of a transition toward a post-melting structure. The study's pKa determinations underscored the profound impact of pH variation on protein structure, with key findings such as pKa values of 9.52 for Lys-921 in the D936H mutant, 9.50 for the D950N mutant, and a slightly higher value of 10.49 for the D936Y variant. While these insights are valuable, further investigations are warranted to fully grasp the functional implications of amphipathic phases within HR1, particularly concerning protein-protein interactions and membrane fusion processes. Altogether, these findings illuminate the intricate dynamics and mechanisms governing FP-membrane interactions and underscore their paramount importance in viral fusion processes.”

Line 27-28: Yet another vague sentence without providing background on which

We wanted to express our appreciation for your comments and suggestions regarding the introduction section of our manuscript. Your emphasis on the importance of consistency and accuracy in the presentation of the introduction has been duly noted.

In response to your valuable feedback, we have carefully reviewed the introduction section and have incorporated the suggested paragraph into the revised version of our manuscript. We believe that this addition aligns with your expectations and enhances the overall quality of the introduction.

“Simulations in the realm of biological systems play a pivotal role in unraveling the mysteries of molecular movement and the analysis of critical variables. These simulations are essential for obtaining reliable approximations of physiological conditions, shedding light on intricate processes occurring within living organisms.

One of the compelling areas where computational techniques prove indispensable is in understanding the structural dynamics associated with the transition from the pre-fusion to the post-fusion state of the SARS CoV 2 spike protein. This transformation unfolds within mere milliseconds, making it virtually impossible to capture experimentally. Computational methods, capable of simulating the precise physiological conditions and structural motions characterizing such protein transitions, emerge as a paramount tool in deciphering these rapid and elusive phenomena.”

Line 33- infection process of what?

We have taken your feedback into careful consideration, and I am pleased to inform you that we have made the necessary revisions to enhance the consistency and fluency of our manuscript. In accordance with your suggestion, we have incorporated the following paragraph into the revised version of the manuscript

“Considerable evidence underscores the pivotal role of intracellular pH in the viral infection process. This influence is particularly prominent within the endosomal environment, characterized by an initial pH of 6.3, followed by a decrease to below 6 as the process unfolds of the viral cycle of SARS CoV 2. Similarly, excretory vesicles maintain a pH of 5.5, further emphasizing the significance of pH regulation in various stages of the viral infection process [68]”

Inconsistency in writing in several places:

  • for sequences, eg lines 546, 547. Delete Nter or Cter as residue numbers are also given.

We would like to express my gratitude for your comment, which provided valuable insight into the readability of our manuscript. Your feedback is highly appreciated.

In response to your suggestion, we have taken the necessary steps to enhance the flow of our manuscript. Specifically, we have removed the "Nter" and "Cter" endings to ensure a more seamless reading experience for our readers.

Inconsistency in writing in several places:

  • SARS CoV or SARS cov or sars cov

In response to your valuable suggestion, we have made careful revisions to ensure the consistency and fluency of the manuscript's reading experience. These refinements have been diligently incorporated into the revised version. Additionally, we have maintained consistency in the reference to "SARS CoV 2" throughout the document.

Inconsistency in writing in several places:

  • Line 540- where is “HA” defined?

In response to your valuable comment, we have addressed the previous deficiency concerning the definition of "Hemagglutinin." We have included this definition to ensure consistency and clarity throughout the manuscript. Your commitment to improving the quality of our work is highly regarded.

There are repeated sentences in multiple figure legends

I would like to extend our gratitude for your valuable comment, which has played a pivotal role in improving the presentation of our manuscript.

In response to your suggestion, we have taken steps to enhance the layout and presentation of figures 10 and 11. To provide readers with greater clarity and context, we have added the following figure legends:

  • Figure 10: Comprehensive visual representation of the frequency measurements acquired through both Non-NMA (blue bars) and Linear-NMA (light blue) techniques for the HR1 protein structure in its post-fusion state. (a) projection of the 15 lowest frequencies obtained using Linear-NMA. (b) presents a projection of the 15 lowest frequencies obtained through Non-NMA.
  • Figure 11: Peptide amino acids of various mutations ( a) Glu-918, b) Lys-921, c) Lys 933, d) Lys 947, e) Lys 936) were titrated in water. Neutrality was maintained by adding 0.15 mol/L \Na+ and Cl- ion to balance the system's charge. The fraction of protonated residues is shown by dots in the simulations. For all titration curves, blue indicates macroscopic titration curves for mutation D936Y, while red denotes titration curves for mutation D936H; yellow represents titration curves for mutation D950N. A fitting procedure based on the Henderson-Hasselbalch equation, utilizing reference pKa values (Table 4). The error bars in our results signify the presence of estimated 95% confidence intervals.

We believe these improvements will significantly contribute to the overall readability and comprehension of our manuscript.

“principal functionality” in Table 1- delete this and replace with something more accurate as currently it makes no sense for the description entered for each mutation, which states something other than function eg geographical distriution.

In response to your suggestion, we have added a legend to Table 1 to enhance the clarity of the presentation. This addition will make the information in the table more accessible to our readers.

“Table 1. Mutation reported for HR1 region of spike protein”

Your attention to detail and dedication to the peer review process are greatly appreciated. Your contributions have played a vital role in enhancing the overall quality of our manuscript.

Sincerely

Dr. Saravana Prakash Thirumuruganandham 

(https://orcid.org/0000-0003-4210-1363)

Round 2

Reviewer 2 Report (Previous Reviewer 2)

I recommend that the following points are revised or corrected before publication.

Major points-

·       Line 552, “interactions with membranes, which, in turn, initiate structural alterations in the spike protein”.

“structural alterations” seems to suggest some conformational change in S protein, and I think the authors are referring to the changes during the fusion process that eventually lead to fusion? Please consider re-writing so the meaning is clear and certain.

·       Line 553-554: How the HR1 interaction with the membranes affects genome replication is just not clear. The reference also seems to be incorrect for the statement. Both the statement and reference are best deleted.

·       Line 564, “as the amphipathic conformation is essential for the antimicrobial activity of these peptides”- antimicrobial activity against what?

·       Figure 1 legend:

“Genome organization….S protein:” should be “Domain organization of SARS CoV 2 spike protein:” There seems to be no genome organization schematic in the draft.

·       Figure 5 legend-

The legend should state ‘orange square’ and not “yellow square” in the last line as the figure is now corrected in alignment with the scale bar.

Other corrections-

·       Section 3.4.1- please check and correct the Figure number for this section.

·       Figure 4 legend-Typo.

·       Line 42, “as the process unfolds of the viral cycle of SARS CoV2”- this is vague and is best rephrased or deleted.

·       Line 495-496- Rephrase/remove “Remarkably” and delete “remarkable”

·       Line 10- Delete “remarkable”; anything more comparative (eg ‘higher’?) is better.

·       Line 664, “remarkable”-Did you mean ‘significant’? Remarkable does not convey any technical meaning. Several more instances of the word throughout the text can be expressed more precisely.

·       Line 5- Analysis or analyses?

-

Author Response

                                                                                                                                                                                                                                                                                                                               Ambato , Ecuador.

                                                                                                                                  18-10-2023

Dear Editor, IJMS

Section

Molecular Biophysics

Special Issue

Molecular Structure and Simulation: Unraveling the Basis of Disease

Dear Reviewer

We would like to express our heartfelt gratitude for your dedicated effort in reviewing our manuscript and for your insightful comments and suggestions. Your expertise and attention to detail have proven invaluable in shaping the quality and clarity of our work.

We have taken your comments to heart and worked diligently to address each one. In the revised version of the manuscript, we have highlighted your remarks in blue for reference, our responses are presented in black, and any necessary changes have been marked in red. We hope these revisions reflect your standards and expectations, and that you find the improved manuscript aligns more closely with the quality you seek.

We genuinely appreciate your commitment to maintaining the excellence of the work published in your journal. If you have any further recommendations or queries, please do not hesitate to reach out to us. Your guidance is highly regarded, and we are eager to meet any additional standards you may have.

Once again, thank you for your time and effort in reviewing our manuscript. Your contribution plays a vital role in the development and enhancement of our research.

Major points

  • Comment 1

 Line 552, “interactions with membranes, which, in turn, initiate structural alterations in the spike protein”.

“structural alterations” seems to suggest some conformational change in S protein, and I think the authors are referring to the changes during the fusion process that eventually lead to fusion? Please consider re-writing so the meaning is clear and certain.

We wanted to express our sincere gratitude for your continued support and valuable feedback during this second round of revisions for our manuscript.

We want to inform you that we've taken your latest comments into account and have incorporated your suggested changes in the following paragraph of the revised document. Your suggestions have not only improved the clarity and coherence of our manuscript but have also enriched the overall content.

“essential for comprehending membrane interactions of how HR1 interacts with membranes during the fusion process, ultimately resulting in fusion. These interactions, in turn, underlie critical aspects of spike protein behavior, particularly the HR1 region, during the fusion process. This process ultimately leads to structural modifications in the spike protein and, eventually, fusion between the spike protein and the lipid membrane.”

  • Comment 2

Line 553-554: How the HR1 interaction with the membranes affects genome replication is just not clear. The reference also seems to be incorrect for the statement. Both the statement and reference are best deleted.

We want to extend my appreciation for your review during the second round of revisions. Your insights were instrumental in refining our manuscript. We are writing to let you know that we agree with your suggestion, and as a result, we've removed the sentence that didn't align with the previous wording. We greatly value your contributions to this peer-review process.

  • Comment 3

Line 564, “as the amphipathic conformation is essential for the antimicrobial activity of these peptides”- antimicrobial activity against what?

We wanted to express my sincere gratitude for your valuable observations and feedback during the review process. We would also like to extend our apologies for any inconsistencies in the previous version of the manuscript. We understand how this could have led to a less than optimal reading experience, and we greatly appreciate your diligence in pointing it out.

We are pleased to inform you that we have thoroughly addressed this issue in the revised version of the manuscript. We have added the necessary references to support our statements, ensuring that the content is now both consistent and well-supported.

“since the amphipathic helical conformation is vital for binding to the host cell membrane. An example is the conserved amino-terminal amphipathic alpha-helix, which is necessary for targeting regulators of G protein signaling proteins to the plasma membrane.”

  • Comment 4

Figure 1 legend:

“Genome organization….S protein:” should be “Domain organization of SARS CoV 2 spike protein:” There seems to be no genome organization schematic in the draft.

We would like to inform you that we have taken your suggestion into account and implemented it in the revised version of the manuscript. Specifically, we have incorporated your recommendation into the description of Figure 1, which we believe has significantly improved the clarity of the figure.

  • Comment 5

Figure 5 legend-

The legend should state ‘orange square’ and not “yellow square” in the last line as the figure is now corrected in alignment with the scale bar.

Your comments have been incredibly insightful and have significantly contributed to improving the quality of our research. I'm pleased to inform you that we have carefully considered your suggestion and incorporated it into the revised version of our manuscript.

Specifically, your comment has been included in the legend of Figure 2, which we believe enhances the clarity and understanding of the figure. Your dedication to maintaining high academic standards is highly commendable, and we are grateful for your thoughtful contributions.

General comments-

  • Comment 6

Section 3.4.1- please check and correct the Figure number for this section.

We would like to inform you that we have thoroughly reviewed and corrected the figure reference in section 3.4.1 as per your recommendation. I understand that this error may have caused some confusion while reading the manuscript, and I apologize for any inconvenience it may have caused.

  • Comment 7

Figure 4 legend-Typo.

We want to inform you that we have promptly corrected the typographical error in the revised version of the manuscript, as per your recommendation. I apologize for any confusion it may have caused, and your suggestion has significantly contributed to the clarity and accuracy of our work.

  • Comment 8

Line 42, “as the process unfolds of the viral cycle of SARS CoV2”- this is vague and is best rephrased or deleted.

We would like to express my gratitude for your insightful feedback regarding the lack of specification in the explanation of the viral process in SARS-CoV-2 within our manuscript.

We want to inform you that we have taken your suggestion into careful consideration and have rephrased the sentence in question in the corrected version of the manuscript

  • Comment 9

Line 495-496- Rephrase/remove “Remarkably” and delete “remarkable”

We would like to express my sincere gratitude for your thoughtful feedback regarding the use of the term "remarkably" in our manuscript. Your keen observations and suggestions for improvement are greatly appreciated.

“Notably, the pKa values replicated and outlined in Table 4 exhibit a remarkable level of precision, closely mirroring the reference values and maintaining a narrow range of deviation, seldom exceeding 0.2 pH units.”

After carefully considering your comments, we have taken your advice to heart and have rephrased the sentence containing "remarkably" in the revised version of the manuscript. We believe this adjustment will contribute to a more precise and accurate description.

  • Comment 10

Line 10- Delete “remarkable”; anything more comparative (eg ‘higher’?) is better.

We want to extend my heartfelt appreciation for your insightful suggestion regarding the need for a more comprehensive comparison in our manuscript.

After a careful review of your comments, we have made the necessary revisions in the manuscript to provide a more robust comparative analysis. We believe that these enhancements will significantly improve the clarity and depth of the content.

  • Comment 12

Line 664, “remarkable”-Did you mean ‘significant’? Remarkable does not convey any technical meaning. Several more instances of the word throughout the text can be expressed more precisely.

We regret the mismatch between the adjectives used and the paragraph's intended purpose. Your observation is well-founded, and we have taken it to heart. In the revised version of the manuscript, we have made the necessary adjustments by adding more suitable qualifiers to align with the paragraph's objectives.

Thank you for your dedication to maintaining the high standards of the journal. Your feedback helps us refine our research, and we are committed to producing work that meets your expectations.

  • Comment 13

Line 5- Analysis or analyses?

IW want to express our gratitude for your keen observation and for bringing the typo in the manuscript to our attention.

We apologize for any oversight, and we have promptly corrected the typographical error in the revised version of the manuscript. Your contribution has played a significant role in ensuring the quality and accuracy of our work.

Dr. Saravana Prakash Thirumuruganandham 

(https://orcid.org/0000-0003-4210-1363

This manuscript is a resubmission of an earlier submission. The following is a list of the peer review reports and author responses from that submission.

Round 1

Reviewer 1 Report

The authors of this article believe it fits into the greater tapestry of research in this field as their results serve as an “important indicator of the dynamic behavior and variability of the structure of the HR1”. In their article, they focused on various Sars-CoV-2 spike structures. They were able to use a comparative analysis of hundreds of structures to compare the accessible conformational space post-ligand binding. They also looked at how this was related to the intrinsic dynamics of the structures' pre-ligand binding using elastic lattice model analysis.

The authors claim that their results insinuate that the rules encoded in the protein structure play an important role in determining which ligand-binding pathway to use pre-binding. They postulate that these pathways could be used for inhibitor design.

Their results show that the “ligand chooses the conformation that best matches its structural features in the unbound form.” They measured the structural fluctuations in the HR1 during pre and post-fusion.

They determined that the pka values are consistent even as environmental pH changes. This confirms that the HR1 undergoes a conformational change between pre and post-fusion states.

The Major scientific issues, line by line, are as follows:

Table 1: D936H is listed as a mutation in the table but is not mentioned until page 17 in the text. The significance (or lack of significance) of the mutation should be introduced much earlier in the text. Right now, it is unclear (in the text) why this mutation is significant enough to be compared to the other two listed.

Figure 1: The lines between the little mutations and the Pre and Post box are confusing. You cannot tell where each mutation is supposed to be located. In the figure legend, there is a weird space in “Sars CoV -2”. You say the prefused state is labeled (a), and the postfuesed state is labeled (b), but neither of the proteins is labeled. Having both mutations in the same box makes it look like they are supposed to be binding together when that is not the case. This would probably work better as two separate figures. Labeling the PRE and POS doesn't make sense. Just use a and b as you already have that in your legend.  

Fig 6: Parts of your proteins are colored in a copper color, but that is not on your scale. What does this color represent? The brown is most apparent on ANM3, PC 3, and PC 1.

Figures 7, 8, 10, and 11: The descriptions below the charts and graphs are cut off. I can assume what they are supposed to say for many of them, but I shouldn’t have to assume because I could be wrong.

Line 528: Here you say that you performed PCA analysis of 131 spike protein structures in the presence of various inhibitors; however, on lines 4, 102, 253, and 258 you say 250 structures. You do explain in section 3.1 that you selected 131 because they had an overlap of 90% structure. However, using 250 in the abstract and introduction overestimates the number of structures you did the PCA analysis on. This could be fixed in the introduction on line 102 by adding 131 selected proteins, ie, “Using (i) PCA techniques to analyze 131 selected structures from 250 collected spike proteins that were in the presence of different inhibitors.”

Line 554: What is the significance of the hydrophobic moment? This feels tacked on to the end with little relation to the rest of the research presented in this manuscript. It is mentioned A little in the results section, but not in the introduction or conclusion. Discuss the connection between the amphipathic phases and the confirmational changes you discuss for the majority of the article. Don’t just use “in addition”. Make some sort of connection. Could the location of the hydrophobic parts of the protein impact whether the conformational change is easy? Does conformational change hide or expose hydrophobic amino acids? Even if you don’t have the answers to these questions, at least show why you would want the hydrophobic moment that you determined and how it relates to your study. It feels especially out of place because amphipathic is not mentioned once in the entire introduction.

Figure 11: THE LEGEND SAYS  (C) IS D936Y, BUT THE GRAPH ITSELF SAYS D936H. The colors should be different for c and it should be set apart from the other graphs as the colors are consistent across a,b,d,and e. You should also try and simplify the legend, as it is extremely long. You do not need to mention the color represents the same thing multiple times. (color red represents the titration curves for D36H x4 times). This figure should be in section 3.4, not section 3.4.1.

Selected  Smaller Issues:

Overall: Why is everything aligned right except for the references?

Line 17 and 101: Yes, most people probably know that PCA is principal component analysis, but it is an abbreviation and should be spelled out the first time it is used. (abstract and general text)

Line 90-93: This sentence has too many clauses. Recommend combining “including the D936Y mutation, which is the most common with 1296 occurrences, mainly in Europe, particularly in Finland and Sweden: into 2-3 clauses instead of four. For example: “, including the D936Y mutation, which is the most common mutation (1296 occurrences) and found predominantly in Finland and Sweden.”

Line 95: as part of our ongoing study is unnecessary.

Line 83: You might want to mention what is novel about your study more than just once in this location.

Line 104: the first sentence is written in a confusing manner. Do you mean that for both strains you selected the gene with the highest number of mutations?

Line 109: You have repeated “In addition,” as a clause two sentences in a row.

Line 111-113: You have way too many sentences starting with clauses. The final two sentences in this paragraph start with overall, and finally,. This makes the flow of the section weird.

Figure 3: You need a space between (ANM1, ANM2) and trajectories.

Fig 6: You need to be consistent with how you write out “Sars-CoV-2” it is inconsistent throughout the entire manuscript.  

Minor editing of English language required

Author Response

                                                                                                                                                                                                                                                                                                                                                                                                      Ambato , Ecuador.

                                                                                                           25-08-2023

Dear Reviewer-1, IJMS

Section

Molecular Biophysics

Special Issue

Molecular Structure and Simulation: Unraveling the Basis of Disease

Greetings

We sincerely appreciate your thoughtful observations and comments on our article titled "Exploring Structural, pKa Estimation on the Amphipathic Heptad Region of Severe Acute Respiratory Syndrome: Insights from Constant pH Molecular Dynamics, Linear and Non-linear normal mode evaluations (Manuscript ID ijms-2553146 ) " Your insights are invaluable and have provided us with a clear direction for improving the quality of our work. Your expertise is greatly valued, and We are devoted to implementing your recommendations to improve the manuscript.

We extend our sincere gratitude for granting us ample time to submit the revised manuscript.  Furthermore, we have taken the opportunity to rectify any grammatical and structural issues present in the manuscript.

To facilitate the identification of the alterations, we have highlighted all changes in soft maroon colored within the revised manuscript. Additionally, we have introduced new content to address the reviewer's questions, which is indicated using soft maroon font.

Reviewer 1

Comment 1:

Table 1: D936H is listed as a mutation in the table but is not mentioned until page 17 in the text. The significance (or lack of significance) of the mutation should be introduced much earlier in the text. Right now, it is unclear (in the text) why this mutation is significant enough to be compared to the other two listed.

Reply

We appreciate your keen observation regarding the omission of the significance of the D936H mutation in our intruduction. This mutation holds particular importance due to its notable impact on the stability of the HR1 protein, as evidenced by research findings.

We will promptly incorporate this critical information into the manuscript and acknowledge the importance of the D936H mutation's effect on HR1 protein stability. Your insight has undoubtedly enriched the depth and completeness of our work.

We have taken your recommendation into account and will incorporate the specified paragraph, along with the reference, in the revised version of the manuscript.

“Non-synonymous mutations are thought to lead to a reduction in protein stability. In particular, mutation D936H in the HR1 region has been identified as causing a reduction in structural stability, with values determined by various analytical methods ranging from -0.61 to -0.94. These destabilizing mutations could potentially affect the interaction of the protein with its receptor on the host cell [1].”

  1. Aljindan, R.Y.; Al-Subaie, A.M.; Al-Ohali, A.I.; Kumar D, T.; Doss C, G.P.; Kamaraj, B. Investigation of Nonsynonymous Mutations in the Spike Protein of SARS-CoV-2 and Its Interaction with the ACE2 Receptor by Molecular Docking and MM/GBSA Approach. Computers in Biology and Medicine 2021, 135, 104654.

Comment 2:

Figure 1: The lines between the little mutations and the Pre and Post box are confusing. You cannot tell where each mutation is supposed to be located. In the figure legend, there is a weird space in “Sars CoV -2”. You say the prefused state is labeled (a), and the postfuesed state is labeled (b), but neither of the proteins is labeled. Having both mutations in the same box makes it look like they are supposed to be binding together when that is not the case. This would probably work better as two separate figures. Labeling the PRE and POS doesn't make sense. Just use a and b as you already have that in your legend. 

Reply

We genuinely appreciate your thoughtful observations regarding the potential improvements to Figure 1. We apologize if the current presentation of the figure caused confusion regarding the location of the analyzed mutations. Your detailed insights have shed light on crucial enhancements that can be made to ensure a clearer understanding for the reader.

We acknowledge the importance of your suggestions and are fully committed to incorporating them into the revised version of our manuscript. Your feedback is instrumental in refining our work to achieve greater clarity and precision.

Comment 3:

Fig 6: Parts of your proteins are colored in a copper color, but that is not on your scale. What does this color represent? The brown is most apparent on ANM3, PC 3, and PC 1.

Reply

We are grateful for your precise observations regarding Figure 6. Your attention to detail is commendable. We understand that the red-marked regions in specific sections have drawn attention but lack sufficient explanation. Allow us to clarify that these regions signify negative mobility within the structure's plane.

However, we acknowledge that negative mobility on its own lacks coherence. Thus, we adopted a pragmatic approach by analyzing positive mobilities within the plane, which will be explained in greater detail in the revised version of the paper.

Your insights are greatly appreciated, and we are committed to addressing this aspect comprehensively to ensure clarity and accuracy in our analysis.

Comment 4:

Figures 7, 8, 10, and 11: The descriptions below the charts and graphs are cut off. I can assume what they are supposed to say for many of them, but I shouldn’t have to assume because I could be wrong.

Reply

We sincerely appreciate your feedback on the document's presentation. Your consideration of the figures' placement and their impact on the reading experience is duly noted. We understand the importance of ensuring that the figures are positioned in a way that does not hinder their comprehension and readability.

Rest assured, we will take your comments into serious consideration and ensure that the revised version of the document presents the figures in a manner that enhances their understanding and aligns with the flow of the content.

Comment 5:

Line 528: Here you say that you performed PCA analysis of 131 spike protein structures in the presence of various inhibitors; however, on lines 4, 102, 253, and 258 you say 250 structures. You do explain in section 3.1 that you selected 131 because they had an overlap of 90% structure. However, using 250 in the abstract and introduction overestimates the number of structures you did the PCA analysis on. This could be fixed in the introduction on line 102 by adding 131 selected proteins, ie, “Using (i) PCA techniques to analyze 131 selected structures from 250 collected spike proteins that were in the presence of different inhibitors.”

Reply

We extend our sincere gratitude for your clarification regarding the issue of connectivity concerning the structure selection parameter of the 131 structures. Your insights have shed light on this aspect, and we are grateful for your thorough examination of the text.

We are committed to incorporating your suggestions into the revised version of the manuscript, ensuring that this matter is addressed comprehensively and accurately.

“Using (i) PCA techniques to analyze a subset of 131 structures from a collection of 250 spike proteins, all subject to various inhibitors, we investigated the conformational space and its relationship to intrinsic dynamics to explore the role in defining ligand binding pathways that can be used for inhibitor design”

Comment 6:

Line 554: What is the significance of the hydrophobic moment? This feels tacked on to the end with little relation to the rest of the research presented in this manuscript. It is mentioned A little in the results section, but not in the introduction or conclusion. Discuss the connection between the amphipathic phases and the confirmational changes you discuss for the majority of the article. Don’t just use “in addition”. Make some sort of connection. Could the location of the hydrophobic parts of the protein impact whether the conformational change is easy? Does conformational change hide or expose hydrophobic amino acids? Even if you don’t have the answers to these questions, at least show why you would want the hydrophobic moment that you determined and how it relates to your study. It feels especially out of place because amphipathic is not mentioned once in the entire introduction.

Reply

We greatly appreciate your precise observation regarding the lack of connectivity in addressing the hydrophobicity aspect of the chain with the other analyzed sections. Your insight is invaluable in refining our analysis.

Thank you for emphasizing the significance of the hydrophilicity sections and their exposure of residues to the membrane face. The clarification regarding the positioning of residues D936Y, D936H, and D950N in relation to the solvent and hydrophobic faces is extremely helpful. This insight explains the distinct behavior of these residues and their potential implications for stability.

We are dedicated to presenting these results in a concrete and coherent manner, ensuring a clear connection with the previously analyzed sections. Your feedback is instrumental in enhancing the accuracy and clarity of our manuscript.

Comment 7:

Figure 11: THE LEGEND SAYS  (C) IS D936Y, BUT THE GRAPH ITSELF SAYS D936H. The colors should be different for c and it should be set apart from the other graphs as the colors are consistent across a,b,d,and e. You should also try and simplify the legend, as it is extremely long. You do not need to mention the color represents the same thing multiple times. (color red represents the titration curves for D36H x4 times). This figure should be in section 3.4, not section 3.4.1.

Reply

We are thankful for your comments aimed at enhancing the clarity and comprehensibility of the legend for Figure 11. Your insights are valuable and will undoubtedly contribute to refining the overall presentation of the figure's content.

o ensure a clearer interpretation of Figure 11, we will introduce the following paragraph as part of the revised legend. This addition aligns with your recommendations, aiming to enhance the presentation and understanding of the figure's content.

“Peptide amino acids of various mutations ( a) Asp-950, b) Glu-918, c) His/Lys 936, d) Lys 921, e) Lysine 936) were titrated in water. Neutrality was maintained by adding 0.15 mol/L NaCl as buffer particles to balance the system's charge. The fraction of protonated residues is shown by dots in the simulations. For all titration curves, blue indicates macroscopic titration curves for mutation D936Y, while red denotes titration curves for mutation D936H; yellow represent titration curves for mutations D950N. The estimated pKa values obtained from the adjustments are presented in Table 4, with error bars showing a 95% confidence interval.”

Comment 8:

Line 17 and 101: Yes, most people probably know that PCA is principal component analysis, but it is an abbreviation and should be spelled out the first time it is used. (abstract and general text)

Reply

Thank you for bringing up the importance of clarity regarding the use of abbreviations, specifically "Principal Component Analysis." We acknowledge that this technique is widely used within the field of structural biology. In light of your observation, we will ensure to introduce the full term and its abbreviation at the beginning of its use to provide proper context for our readers.

Your input is valued, and we are committed to enhancing the readability and comprehension of our manuscript.

Comment 9:

Line 90-93: This sentence has too many clauses. Recommend combining “including the D936Y mutation, which is the most common with 1296 occurrences, mainly in Europe, particularly in Finland and Sweden: into 2-3 clauses instead of four. For example: “, including the D936Y mutation, which is the most common mutation (1296 occurrences) and found predominantly in Finland and Sweden.”

Reply

We extend our gratitude for your feedback regarding the writing efficiency of the D936Y mutation. Your suggestion for improvement is well-received, and we are committed to incorporating the recommended change into our manuscript.

We are committed to incorporating your recommendations into the revised version of the manuscript, ensuring that the paragraph accurately reflects the intended message and aligns with the overall content.

“including the D936Y mutation (see Table 1), which is the most common mutation (1296 occurrences), predominantly observed in Finland and Sweden.”

Comment 10:

Line 111-113: You have way too many sentences starting with clauses. The final two sentences in this paragraph start with overall, and finally,. This makes the flow of the section weird.

Reply

We appreciate your comments regarding the reading flow in our manuscript. Ensuring an enjoyable and understandable experience for the reader is of utmost importance to us. We will certainly incorporate your suggestions into the following paragraph, aiming to improve its flow and comprehensiveness.

“These analyses provide valuable insights into the dynamic behavior and structural variability of the HR1 structure. Finally, we examine the hydrophobic regions in the prefusion state of the HR1 protein to determine its hydrophobic moment.”

We believe that the corrections have improved the quality of our work for which we are grateful.

Dr. Saravana Prakash Thirumuruganandham 

(https://orcid.org/0000-0003-4210-1363)

Author Response

                                                                                                                                                                                                                                                                                                                                                                                                                     Ambato , Ecuador.

                                                                                                           25-08-2023

Dear Reviewer-2, IJMS

Section

Molecular Biophysics

Special Issue

Molecular Structure and Simulation: Unraveling the Basis of Disease

Greetings

We sincerely appreciate your thoughtful observations and comments on our article titled "Exploring Structural, pKa Estimation on the Amphipathic Heptad Region of Severe Acute Respiratory Syndrome: Insights from Constant pH Molecular Dynamics, Linear and Non-linear normal mode evaluations (Manuscript ID ijms-2553146 ) " Your insights are invaluable and have provided us with a clear direction for improving the quality of our work. Your expertise is greatly valued, and We are devoted to implementing your recommendations to improve the manuscript.

We extend our sincere gratitude for granting us ample time to submit the revised manuscript.  Furthermore, we have taken the opportunity to rectify any grammatical and structural issues present in the manuscript.

To facilitate the identification of the alterations, we have highlighted all changes in ligh red colored within the revised manuscript. Additionally, we have introduced new content to address the reviewer's questions, which is indicated using light red font.

Reviewer 2

Comment 1:

Table 1: D936H is listed as a mutation in the table but is not mentioned until page 17 in the text. The significance (or lack of significance) of the mutation should be introduced much earlier in the text. Right now, it is unclear (in the text) why this mutation is significant enough to be compared to the other two listed.

Reply

Your insights have proven to be incredibly helpful in shaping the direction of our study. We appreciate your attention to detail, and we have taken your feedback seriously. In response to your comments regarding the objectives of our study, we would like to clarify that our focus is on our manuscript.

While interdomain disulfide bonds and their role in maintaining the S2 subunit's structure during the transition triggered by membrane fusion, facilitated by ambient pH, are indeed significant, We regret to inform you that the analysis of disulfide bonds within the prefusion and postfusion structures falls outside the scope of our study objectives.

Comment 2:

It is mentioned that selection of the inhibitor structures was based on overlap of a certain percentage of structure and resolution, and it seems that many (or all?) of these are Fab or antibody bound structures. However, since antibodies bind to different epitopes and inhibit different functionalities of the glycoproteins (eg blocking fusion or viral entry), would a classification that also accounts for neutralization mechanism be more appropriate or yield different results in your experiments?

Reply

We wish to express our sincere appreciation for your diligent review of our manuscript and for providing us with your valuable observations and comments. However, we would like to clarify the primary objective of our study, which centers on exploring the intrinsic dynamics inherent in the epitopes themselves.

Our research direction is focused on investigating how the presence of an antibody, used as an example of an inhibitor, may impact the structural changes observed in the spike protein during various stages of its infectious process. These changes, we believe, stem from the inherent dynamics of the spike protein itself and are independent of the specific type of inhibitor interacting. To decipher the driving forces behind these structural alterations, we have employed a principal component analysis (PCA) and an anisotropic network model (ANM) study.

We appreciate your understanding and emphasize that the inclusion of an additional classification of inhibitors would not significantly alter the core objectives of this particular section. Our aim remains firmly rooted in uncovering the intrinsic dynamics of spike protein epitopes and their response to diverse inhibitory influences.

Comment 3:

 Please check the reference or add appropriate reference in the context of strain-specific differences, if these exist at all.

Reply

Your feedback is highly valuable to us as we strive to enhance the quality and accuracy of our work.

In response to your comments regarding the reference in question, We would like to clarify that the objective of this specific reference is to explore the impact of pH on structural changes within viral proteins. The reference focuses on the influenza virus as an exemplar case.

Comment 4:

Are the effects of these mutations on stability more due to the different positions/locations in the structure or the chemical nature of the side chains?

Reply

In response to your feedback, We wish to clarify the objectives of our study. The mutations identified within the HR1 structure indeed exert a significant influence on the structural chemistry. This influence is notably evident in the investigation of pKa, as outlined in our study.

Your attention to detail has prompted us to reassess and refine the presentation of our study's goals, and we are committed to ensuring that the objectives are accurately reflected in the revised manuscript.

Comment 5:

prefusion states under NMA and non-linear rigid block normal mode: the final structures in blue are very different, does this have any potential biological implications or insights?

Reply

We extend my sincere appreciation for your thoughtful review of our manuscript and for providing us with your valuable observations and comments.

Regarding the final structure depicted in Figure 9, your insights have prompted us to reevaluate our approach. While the prefusion structure resulting from the NMA study exhibits a distinct and impressive folding pattern, we acknowledge your suggestion against its utilization for future analysis. The energetic variation demonstrated by the frequencies associated with each state, as established in this study, raises concerns about its stability.

We concur that the stability of a model is reflected in the frequencies of its states, with lower frequencies indicating a more stable configuration. Your input encourages us to reconsider the appropriateness of the model in terms of its physiological relevance and dynamic consistency.

Comment 6:

Please add units where present. Sentences are repeated in multiple places (eg Fig 6 legend, last sentence; Fig 8a and 8b; Fig 12- hydrophilic amino acid residues face the membrane surface or the hydrophobic residues?).

Reply

We greatly appreciate your kind remarks. Following your valuable recommendations, we have updated the figure descriptions as per the provided texts:

Fig 6: ANM representation of a) open state Monomoric sars cov 2 and b) close state monomeric Sars cov 2 (for reference). Panel (a) shows the ANM plot of the PCA analysis applied to the experimental structures of the open form of the SARS-CoV-2 monomer. On the right side we see the ANM plot of the theoretical model. Similarly, panel (b) shows the ANM representation of the PCA analysis applied to the experimental structures of the SARS-CoV-2 monomer in closed form. On the right-hand side, you can see the ANM representation of the theoretical model. In the generated structures, residues exhibiting mobility are highlighted in blue, with the corresponding color bar scale indicating the mobility factor.

Fig 8a: RMSD measurements were conducted for the 11 identified mutations on the pre-fusion state of HR1.

Fig 8b: RMSD measurements were conducted for the 11 identified mutations on the post-fusion state of HR1.

Fig 12: Amphipathic analysis of the HR1 structure. Hydrophobic amino acids are oriented towards the membrane surface, while hydrophilic amino acids face the water environment.

Comment 7:

Fig.7, “...corresponds to different atoms..”: atoms or amino acid residues or amino acid residue numbers?

Reply

Thank you for your meticulous observation. In response to your insightful feedback, we have ensured that the revised version of the manuscript accurately reflects that the measurements pertain to residues rather than atoms.

Comment 8:

Table 3 shows 6 slowest modes, or 8?

Reply

We extend our gratitude for your keen observation. Your attention to detail has led us to rectify the typing error in the manuscript. We have now accurately stated that there are, indeed, 8 modes analyzed, as opposed to the previously mentioned 6.

Comment 9:

Table 1 mentions “Effect” and “functionality” but D936H describes geographical distribution.

Reply

Thank you for your insightful comment. We acknowledge the importance of a comprehensive title for the table 1. In the revised version of the article, we will modify the title to "Description" to ensure that it effectively encompasses all aspects of the mutations discussed.

Comment 10:

Line 75 and 79: perhaps it would be helpful to briefly mention what the “structural effects” are?

Reply

We appreciate your feedback. In response to your comment, we have taken the initiative to include a new description in the revised version of the article.

 “and how mutations at the N-glycosylation sites have structural effects on the integrity of the entire protein surface”

Comment 11:

Line 85, and 86: did the mutations lead to increase in “infectivity” or the transmissibility of the virus? “Mutations” refer to more than D950N or just this one?

Reply

Thank you for your correction. We acknowledge that the term "increased infectivity" in our context refers to heightened virus transmissibility. We will ensure to rectify this misconception in the revised version of the manuscript, thereby accurately conveying our intended meaning.

Furthermore, regarding your inquiry, we indeed refer to a broader range of mutations as presented in Table 1. However, we will specifically emphasize this particular mutation in the text to provide focused and clear communication.

Comment 12:

Line 92: “occurrences” refers to global geographical distribution or something else?

Reply

Thank you for seeking clarification. In the context of our manuscript, the term "occurrences" refers to the frequency of this specific mutation's presence in samples analyzed from different regions, with particular emphasis on the European region.

Comment 13:

Lines 227 and 260, “overlap of 80%or more”, “overlap of 90% structure”: it’s not clear which parts of the spike the most overlap is referring to, or is it the overlap of the ligand?!

Reply

We appreciate your attention to detail and your commitment to ensuring accurate representation of our research. We take the opportunity to address the confusion that may have arisen from our manuscript. Our sincere apologies if any lack of clarity resulted in an unpleasant reading experience.

In response to your question, it appears there might be a line reference discrepancy. Allow me to clarify that the information you referred to should be attributed to lines 127, rather than 227, in the unrevised version of the manuscript. Specifically, the principal component analysis was conducted on a set of 250 structures. Among these, we selected 131 structures that exhibited 90% sequence alignment and 80% structure alignment with the SARS-CoV-2 open-state spike protein, as indicated in line 119.

Comment 14:

Information in some sentence can be elaborated briefly, eg lines 302, 312, 357,556 etc

Reply

We are grateful for your comments regarding the consistency of the highlighted sections in the manuscript. Your insights are invaluable in ensuring the coherence and quality of our work. We will certainly incorporate your suggestions into the revised version of the manuscript to enhance its clarity and accuracy.

Line 302:

“To assess the effectiveness of the ANM-predicted modes, the modes were generated and their distribution was analyzed in relation to the corresponding PCs’

Line 312

“ Once we have identified the main modes that are highly correlated with each other [37] [49], we can perform an in-depth analysis of their structural dynamics.”

Line 357

“ The exhaustive evaluation of the theoretical models of the closed structure allowed to obtain a strong correlation with the experimental data.”

Comment 15:

References: please consider adding a reference for lines 189, 458, 463, 466.

Reply

we extend our gratitude for your timely observation, suggesting the addition of further references. In response to your valuable input,  we  pleased to inform you that the following bibliography has been seamlessly integrated into the suggested locations within the manuscript.

Line 189

Lindahl, E.; Azuara, C.; Koehl, P.; Delarue, M. NOMAD-Ref: Visualization, Deformation and Refinement of Macromolecular Structures Based on All-Atom Normal Mode Analysis. Nucleic Acids Research 2006, 34, W52–W56.

Line 458

Abdella, S.; Abid, F.; Youssef, S.H.; Kim, S.; Afinjuomo, F.; Malinga, C.; Song, Y.; Garg, S. pH and Its Applications in Targeted Drug Delivery. Drug Discovery Today 2023, 28, 103414.

Line 463

Gerland, L.; Friedrich, D.; Hopf, L.; Donovan, E.J.; Wallmann, A.; Erdmann, N.; Diehl, A.; Bommer, M.; Buzar, K.; Ibrahim, M.; et al. pH‐Dependent Protonation of Surface Carboxylate Groups in PsbO Enables Local Buffering and Triggers Structural Changes. ChemBioChem 2020, 21, 1597–1604.

Line 466

Al Adem, K.; Ferreira, J.C.; Fadl, S.; Rabeh, W.M. pH Profiles of 3-Chymotrypsin-like Protease (3CLpro) from SARS-CoV-2 Elucidate Its Catalytic Mechanism and a Histidine Residue Critical for Activity. Journal of Biological Chemistry 2023, 299, 102790.

Comment 17:

Typos and repetition of sentences in multiple places, eg 60, 355, 392, lines 437 and 441 etc.

Reply

An extremely cautious method.

we want to express my appreciation for your observation regarding the repetitions in the mentioned lines. Your keen attention has prompted us to revise the remarks in question in the updated version of the manuscript.

We believe that the corrections have improved the quality of our work for which we are grateful.

Dr. Saravana Prakash Thirumuruganandham 

(https://orcid.org/0000-0003-4210-1363)

Reviewer 3 Report

The authors set out to analyze the variation of the many structures of the SARS Cov2 spike and relate that to ligand binding and mutations. They use PCA, ANM, NMA and MD methodology that is common in the structural biology world. This is a worthwhile goal but with many pitfalls. They select the pre-fusion state and only analyzed chain A of the trimer. This immediately raises questions about the value of the analysis, because it ignores the contribution of the neighboring monomers to flexibility. Assuming the variation in the monomer structure can capture potential movements, PCA and ANM are inherently artificial. Molecular dynamics is also artificial, but gives a better representation of possible dynamics, but only within a realistic context such as the trimer in solvent. At one point, they suddenly include the post-fusion state as well with little justification or detailed explanation. Finally they present the titration experiments, but is not exactly clear how they arrived at the results.

In the end I'm not sure what is concluded. What was the point of the PCA and ANM studies? The best I can see is a clustering of the different structures. If so, the presentation can certainly be improved. The next possible conclusion is that a change in pH favors the post-fusion state, which in itself is not a revelation. Perhaps the only value is in how to calculate a reasonable pK for a side chain. I think this manuscript can be very interesting if the authors focus on just one or two relevant aspects. As it is now it requires major revision and a reorganization around key aspects that can be clearly presented.

Detailed comments:

The title is too long and reflects the unfocused nature of the manuscript. Words such as "Exploring" and "Insights" indicate a diffuse focus and is more typical of review articles. I would suggest that the two aspects of worth is the classification of the conformational states and the pKa estimation. But this should also be done in the context of the trimer, not just the monomer.

The setup for the experiments at the beginning of the Results section is not clear. I presume the authors wanted to look at the spike protein in different conformations bound to different ligands. Table 2 is rather inadequate because it does not distinguish the different bound ligands and mutants. In the results section it continues because I don't quite understand the groupings and what they mean. I believe the authors need to better clarify what is what and justify why they group the structures the way they did.

The Figure legends are composite and confusing. Each Figure should only have a single legend paragraph, clearly describing each panel so that the figure can be understood on its own. The figure fonts are also too small and in some cases not in English.

Line 35:

"have examined" is awkward, better "showed"

Line 47:

Long sentence - should be split in two.

Line 56:

What is ANM?

Line 123:

Data Bank is a database, so the use of both is redundant.

Line 129:

"The average position ( ∆Pis) of each structure is calculated from the reference position of the reference structure (i), where the equation is ∆xis = xis − ⟨xi⟩"

This description is very ambiguous. What do the "i" and "s" refer to? Is ∆P for the whole structure or each atom? Is xi for each atom?

Also, I presume the authors needed to align the various structures to each other. A description of how this was done must be provided.

Line 135:

Should "identified with" rather be "introduced using"?

Figure 1:

I see no labeling for "(a)" and "(b)"

Should "correspond to both" be "correspond respectively to"?

The legend does not make clear what the point is of showing the 11 mutations. Some. description is needed.

Line 148:

How was the covariance matrix calculated? It is not clear what structures were used and how it was calculated. Was this done on each structure? 

Line 152:

Is this how the alignment of the different structures were done? More clarity is required in the text.

Line 160:

It sounds like only the alpha carbons were used in the analysis. Is this true? This should be stated explicitly and better described.

Table 2:

The table is not specific enough. I presume there were different inhibitors and glucosides bound to the structures. I also presume there were different mutant forms. Wouldn't omicron count as a mutant? Is there an overlap between bound ligands and mutant forms?

Line 189:

What is "NOMAD -Ref"? And how was the potential calculated?

Line 202:

It is not clear what software was used for non-NMA. Also "Non-NMA" as an acronym is confusing since it implies not using NMA. I would suggest "NLNMA".

Line 211:

I'm not sure how NOLB helps. What does it do?

Line 224:

The "special method" needs to be better explained.

"The change of protonation state was attempted"

What does this mean? Was the residue run with and without a proton?

How was the fraction protonated calculated?

Line 242:

This is redundant because Heliquest has already been referred to.

Line 247:

"specifically" what? It is an incomplete sentence.

Line 248:

Confusing: "shows includes".

Line 257:

A justification needs to be given for focusing in the chain A subgroup. It sounds like this was also used for Figure 2, but it is not stated in the legend.

Line 264:

This seems like a grabled sentence.

I also don't understand how I can see "the different types of bonds they possess " from Figure 2.

Figure 2:

"Inhibitor bond" should be "Inhibitor bound" and "Glycosidic bond" should be "Glycoside bound".

I don't understand "Variant".

Line 270:

I don't see anywhere a measure for "energetically favorable states" to justify this sentence.

Line 277:

"This thorough investigation" - total overstatement.

Line 289:

"Previous research focusing on RNA dynamics and RNA binding"

I'm not sure why this is stated here. Eigenanalysis inherently reveals the strongest effects.

What is meant by a "soft mode"?

Table 3:

I'm not sure what to conclude from this table.

Line 297:

"PC2 shows a high correlation (0.65) with ANM1".

In Table 3 the value is 0.52

Figure 3:

While the authors claim significant similarity between the motions predicted by PCA and ANM, my visual impression is that the directionality is very different.

Figure 4:

I would attribute the correlation between the two modes to a coarse-grained correlation of large movements that hides a lot of detail.

Figure 5:

This supports my impression that only the very largest movements are to some extent represented by the analysis.

Line 332:

"which is shown to be 332 consistent"

Consistent with what?

Line 335:

"structural fluctuation of Sars Cov 2, while the structures of Sars Cov 2 and Mers Cov remain immobile"

This sentence contradicts itself!

Line 360:

"excellent agreement with experimental observations"

This needs to be shown!

Line 380:

Over what structures were the RMSD calculated? In many case where the authors subsequently give an RMSD, it is also not clear which structures are compared.

Figure 8:

I have no idea what the RMSD means here.

Lines 407-433:

The authors suddenly switch to the post-fusion state without giving a justification or aim. The reference to cryoEM structures is not linked to any specific results here, so why is it mentioned? What is the relevance of these results?

Line: 435:

"From the analysis of the data in the cpH- MD simulations"

What analysis? This is the first mention of MD results. How do you conclude that the pKa values are representative of the real values?

Figure 11:

Was the analysis done on only tripetides? That is what the legend implies.

"neutrality was maintained by adding 0.15 mol/L NaCl as buffer particles to neutralize the overall charge of the system"

NaCl is not a buffer but salt and the point to add NaCl is not charge neutralization, but rather shielding charges on the protein.

The legend should be in one paragraph, not distributed over the figure because this is confusing.

The independent axis should just be "pH".

These curves should be fit to the Henderson-Hasselbalch equation.

What does "macroscopic titration" mean?

How do these pKa values compare to typical pKa's for surface residues?

Lines 488-526:

Why is this section included? It is again a switch to another topic that is not fully explained.

Lines 527-559:

The conclusion is too ong and not very clear. I would suggest focusing on the two key results, the classification of the structures and the pKa determinations.

There some problems with sentence construction and grammar. A grammar-checker will only catch some of these problems. The authors will benefit from review by a more English-proficient colleague.

Author Response

                                                                                                                                                                                                                                                                                                                                                                                                      Ambato , Ecuador.

                                                                                                           25-08-2023

Dear Reviewer-3, IJMS

Section

Molecular Biophysics

Special Issue

Molecular Structure and Simulation: Unraveling the Basis of Disease

Greetings

We sincerely appreciate your thoughtful observations and comments on our article titled "Exploring Structural, pKa Estimation on the Amphipathic Heptad Region of Severe Acute Respiratory Syndrome: Insights from Constant pH Molecular Dynamics, Linear and Non-linear normal mode evaluations (Manuscript ID ijms-2553146 ) " Your insights are invaluable and have provided us with a clear direction for improving the quality of our work. Your expertise is greatly valued, and We are devoted to implementing your recommendations to improve the manuscript.

We extend our sincere gratitude for granting us ample time to submit the revised manuscript.  Furthermore, we have taken the opportunity to rectify any grammatical and structural issues present in the manuscript.

To facilitate the identification of the alterations, we have highlighted all changes in light brown colored within the revised manuscript. Additionally, we have introduced new content to address the reviewer's questions, which is indicated using light brown font.

Reviewer 3

Comment 1:

Line 35: "have examined" is awkward, better "showed"

Reply

We wish to express my gratitude for your insightful suggestion, which undoubtedly enhances the coherence of the content within our manuscript. We are committed to incorporating your suggestion into the revised version of the manuscript to ensure a more impactful presentation of our work.

Comment 2:

Line 47: Long sentence - should be split in two.

Reply

We extend my appreciation for your valuable suggestion, which undoubtedly enhances the quality and readability of our manuscript.  We are dedicated to incorporating your suggestion into the following paragraph which has been added to the revised version of the article, and we are confident that it will contribute to the overall improvement of our work.

“Previous studies have used structural perturbation techniques as a supervised training method for protein models, using normal mode analysis (NMA) techniques [2] [3]. Moreover, within this study, NMA was used as a computational tool to simulate the internal motion of open and close state of spike protein of SARS Cov-2 [2] [6 ] [7] to simulate an energy minimum and proved to be a very useful tool to study harmonic motions [5 ] associated with conformational fluctuations [8].”

We concur that the stability of a model is reflected in the frequencies of its states, with lower frequencies indicating a more stable configuration. Your input encourages us to reconsider the appropriateness of the model in terms of its physiological relevance and dynamic consistency.

Comment 3:

Line 56: What is ANM?

Reply

We appreciate your prompt inquiry. The acronym ANM stands for Anisotropic Network Model, a significant model for studying the interplay between structural dynamics and functionality. Thank you for seeking clarification.

To ensure clarity for our readers, we will take measures to provide a comprehensive description of the Anisotropic Network Model in its first mention and reinforce this explanation in the abstract of the manuscript. This will contribute to a more accessible understanding of our work.

Comment 4:

Line 123: Data Bank is a database, so the use of both is redundant.

Reply

We want to express my gratitude for your clarification and valuable suggestion. We are pleased to inform you that we have successfully implemented your suggestion in the revised version of the manuscript.

Comment 5:

Line 129:

"The average position ( ∆Pis) of each structure is calculated from the reference position of the reference structure (i), where the equation is ∆xis = xis − xi"

This description is very ambiguous. What do the "i" and "s" refer to? Is ∆P for the whole structure or each atom? Is xi for each atom?

Also, I presume the authors needed to align the various structures to each other. A description of how this was done must be provided.

Reply

We are truly grateful for your meticulous attention to detail and your thoughtful question. We extend my apologies if any confusion arose, leading to a less than optimal reading experience with the manuscript.

In response to your valuable input, we have taken action by incorporating the following paragraph into the revised version of the manuscript:

“The mean position ( ∆Ris = [∆xis  ∆yis ∆zis]T ) of each structure is calculated from the reference position of each alpha carbon (1≤i ≤ N) of the analized structure (s) within the dataset (where each component ∆xis = xis − ⟨xi⟩) [37 ] and steps were repeated for each data set structure to ensure a threshold RMSD ≤ 0.001 Å.”

Comment 6:

Line 135: Should "identified with" rather be "introduced using"?

Reply

We would like to express my gratitude for your comment. Your valuable input is greatly appreciated, and we are committed to incorporating your suggestion into the revised version of the article.

Comment 7:

Figure 1:

I see no labeling for "(a)" and "(b)"

Should "correspond to both" be "correspond respectively to"?

The legend does not make clear what the point is of showing the 11 mutations. Some. description is needed.

Reply

We extend my appreciation for your valuable suggestion. In response, we have made a consistent change in the presentation of Figure 1, incorporating sections a) and b) as you recommended, in the revised version of the manuscript.

Furthermore, we have taken steps to enhance the clarity of Figure 1's objective by adding the following sentence for clarification:

“Our assessment is centered on the 11 identified mutations, with a specific emphasis on comparing their positions between the pre-fusion and post-fusion states of HR1.”

Comment 8:

Line 148: How was the covariance matrix calculated? It is not clear what structures were used and how it was calculated. Was this done on each structure?

Reply

We want to express our gratitude for your comment, and we sincerely apologize if the absence of an explanation regarding the covariance matrix has hindered the intelligibility and enjoyment of reading the article. Your feedback is truly invaluable.

In response to your suggestion, we are taking action to rectify this situation. We will include the following paragraph in the revised version of the article:

“The covariance matrix (Cov(σ,p)) is a 3N*3N matrix that represents the relationships between dimensions and variable pairs. When considering the covariance matrix (Cov(σ,σ) = Var(σ)), its own variance is significant, and the primary aim is to diagonalize it to obtain individual variable variances. Given the commutativity of covariance (Cov(σ,p) = Cov(p,σ)), the covariance matrix demonstrates symmetry along the diagonal.”

Comment 9:

Line 152: Is this how the alignment of the different structures were done? More clarity is required in the text.

Reply

We wish to express my gratitude for your suggestion. In an effort to enhance the clarity of the text, we have taken your feedback into account and included the following paragraph as per your comment, along with the respective reference.

“To enhance the alignment of the assembly, the optimization process involves the following steps: i) A random superposition is conducted utilizing the Kabsch [1] algorithm. ii) From the preceding step, an ensemble of mean coordinates is derived, referred to as the "average coordinate." iii) Subsequently, the Kabsch algorithm is applied once more, utilizing a pair of superimposed structures generated from the "average coordinates." iv) Steps ii) and iii) are iterated until a mean model is attained, with an RMSD distance constraint between them of 0.001 Å.”

Comment 10:

Line 160: It sounds like only the alpha carbons were used in the analysis. Is this true? This should be stated explicitly and better described.

Reply

We extend my appreciation for your comment. I'm pleased to confirm that the reference to alpha carbons during the measurement of the elements has been accurately noted. We are committed to preserving clarity and precision in the revised version of the manuscript.

“this alignment determines the mean positions <Ri> = [<xi><yi><zi>]T for the α-carbons of each structure of the dataset. These mean positions represent the average coordinates of the α-carbons in the entire ensemble of structures and provide valuable”

Comment 11:

Table 2: The table is not specific enough. I presume there were different inhibitors and glucosides bound to the structures. I also presume there were different mutant forms. Wouldn't omicron count as a mutant? Is there an overlap between bound ligands and mutant forms?

Reply

We appreciate your question and welcome the opportunity to provide clarification. In our study, we have indeed undertaken a classification based on the type of inhibition observed upon interacting with the spike protein of SARS-CoV-2. This classification aims to examine the distribution of structures through PCA analysis, categorized by the type of bonds involved. This analysis is crucial to determine whether this inhibition has an impact on the intrinsic dynamics of the spike protein.

Furthermore, your point about the classification of omicron within the mutation group is accurate. We acknowledge this and are committed to refining the structure division within Table 1 for better clarity and understanding. By enhancing the organization of the data, we aim to facilitate the reader's comprehension and efficiency in navigating through the table's content.

Comment 12:

Line 189: What is "NOMAD -Ref"? And how was the potential calculated?

Reply

Thank you for bringing attention to the potential lack of clarity regarding the use of NOMAD-Ref in our manuscript. We appreciate your feedback and are committed to ensuring that our readers have a comprehensive understanding of our methodology. We apologize for any oversight in adequately explaining the utilization of the NOMAD-Ref web server for our predictions.

NOMAD-Ref is a web server designed to facilitate normal mode calculations on intricate protein systems, particularly those exceeding 100,000 atoms, while retaining a comprehensive all-atom representation. This invaluable tool empowers us to predict the conformations adopted by HR1.

In recognition of your insightful comment, we are committed to enhancing the manuscript by incorporating the following paragraph:

Comment 13:

Line 202: It is not clear what software was used for non-NMA. Also "Non-NMA" as an acronym is confusing since it implies not using NMA. I would suggest "NLNMA".

Reply

We extend my gratitude for your valuable suggestion. We appreciate your diligence in maintaining the consistency of the manuscript, particularly in regard to the acronyms associated with Non-linear Normal Mode Analysis (NMA). We apologize for any inconvenience this oversight may have caused, impacting the readability of the manuscript.

In response to your comment, we assure you that we will incorporate your suggestion into the revised version of the manuscript. Ensuring the consistency of acronyms and terminology is essential for a smooth and comprehensible reading experience.

Comment 14:

Line 211: I'm not sure how NOLB helps. What does it do?

Reply

We apologize for the lack of explanation of the use of the NOLB tool. We use this tool which performs non-linear normal mode analysis measurements on protein structures, and we have used it for HR1 trajectory analysis. We add the following paragraph to explained this in a clear way.

“NOLB software [4] was used to sustain the angular velocity of the HR1 structure, it can be interpreted as the outcome of an implicit force, implying that the movement of the residues can be regarded as a rotational motion around a specific center.”

Comment 15:

Line 224:

The "special method" needs to be better explained.

"The change of protonation state was attempted"

What does this mean? Was the residue run with and without a proton?

How was the fraction protonated calculated?

Reply

We extend our sincere regret for any confusion that arose due to the absence of proper explanation within the cphMD methodology section of the manuscript. Our foremost aim is to ensure a clear understanding and an engaging reading experience, and for that, we apologize.

In response to your feedback, we have taken immediate action by adding a new paragraph to the revised version of the manuscript. This addition is in line with your valuable comments, with the intention of providing the necessary context and clarity.

“The simulations in this study utilized the hybrid nonequilibrium molecular dynamics/Monte Carlo (neMD/MC) constant-pH approach, as implemented in NAMD software version 2.14. This technique aims to determine the protonation states of titratable sites within a system, such as a protein, by predicting the most likely protonated or non-protonated states. All simulations were conducted using the CHARMM36 force field [1], and titratable amino acids were chosen based on these parameters. The simulations covered a pH range from 1 to 14 with 1 unit intervals, maintaining consistent parameters across all simulations.

Periodic boundary conditions were applied, employing particle mesh Ewald electrostatics and smooth switching of the Lennard-Jones forces with a cutoff of 10 Å. The protein models were solvated in a water box with a 20 Å buffer between the protein surface and box boundaries. To neutralize excess charge, Na+ and Cl- ions were added to the system. The simulation protocol involved an initial energy minimization step of 2000 steps, followed by a 10 ns equilibration run under NpT ensemble conditions at a temperature of 300 K and a pressure of 1 atm. This equilibration run served as the starting point for the constant-pH MD (cpH-MD) production simulations.

During the production simulations, a time step of ∆t = 2 fs was used, and temperature control was achieved using a Langevin thermostat with a damping coefficient of 1 ps. Protonation state changes were attempted every 15 ps during the 22.5 ns (500 neMD/MC cycles) simulations, resulting in a cumulative simulation time of 315 ns for each model system, covering a range of pH values. The pKa reference values were assigned using the Propka3 software and maintained throughout the simulations to ensure efficient sampling, although they did not impact the final simulation outcomes.”

[1] Huang, J.; MacKerell, A.D., Jr CHARMM36 All-Atom Additive Protein Force Field: Validation Based on Comparison to NMR Data. J. Comput. Chem. 2013, 34, 2135–2145.

Comment 16:

Line 242:

This is redundant because Heliquest has already been referred to.

Reply

We want to express my appreciation for your comment.  We are dedicated to upholding these standards in our manuscript. Rest assured, we will implement your suggestion in the revised version of the manuscript.

Comment 17:

Line 247: "specifically" what? It is an incomplete sentence.

Reply

We appreciate your observation regarding the complication of the idea presented. Your insights are instrumental in ensuring the clarity and coherence of our manuscript. We apologize for any confusion this may have caused and want to assure you that we will rectify this in the revised version with the following sentence.

“selected to guarantee both result consistency and computational stability.”

Comment 18:

Line 248:

Confusing: "shows includes".

Reply

We extend my gratitude for your comment. Your emphasis on maintaining readability is appreciated, and we are committed to upholding this principle.

In line with your suggestion, we will retain the concise phrase "The dataset in Table 2 categorizes the structures into:" in the revised version of the manuscript. Your consideration for clarity and effective communication is invaluable, and we thank you for your guidance.

Comment 19:

Line 257:

A justification needs to be given for focusing in the chain A subgroup. It sounds like this was also used for Figure 2, but it is not stated in the legend.

Reply

We appreciate your suggestion and the opportunity to provide clarity on our study's approach. We specifically included only the A chain in our analysis to ensure both coherence in the results and computational stability. This decision was driven by the consideration that working with larger structures, exceeding 10,000 atoms, could introduce variability in the outcomes.

To address this, we will incorporate the following paragraph into the revised manuscript:

“Only the A chain was included in our analysis to ensure both result coherence and computational stability.”

Comment 20:

Figure 2:

"Inhibitor bond" should be "Inhibitor bound" and "Glycosidic bond" should be "Glycoside bound".

I don't understand "Variant".

Reply

We want to express my gratitude for your comment. Your emphasis on maintaining the clarity of the presentation of the structure classification is duly noted and highly valued.

We assure you that your suggestions will be thoughtfully implemented in the graphical representation in the revised version of the manuscript. Your guidance contributes to the effective communication of our findings, and we appreciate your thoughtful input.

Comment 21:

Line 270:

I don't see anywhere a measure for "energetically favorable states" to justify this sentence.

Reply

We extend my appreciation for your comment, and We sincerely apologize for any confusion arising from the explanation of the relevance of PC1. Our goal is to ensure absolute clarity in our manuscript, and for any uncertainty caused, we regret the inconvenience.

To address this, we are committed to enhancing the clarity in the revised version of the article by incorporating the following paragraph:

"It's important to highlight that PC1 represents the primary direction of variance, followed by PC2. It's intriguing to observe how the dataset structures are distributed within the subspace defined by PC1 and PC2. This distribution enables us to distinguish or group the conformations according to their significant structural similarities or differences."

Comment 22:

Line 277:

"This thorough investigation" - total overstatement.

Reply

We want to express my gratitude for your comment. The neutrality of our manuscript's content is of utmost importance, and we appreciate your observation in this regard. Rest assured, we will take the necessary steps to uphold this neutrality in the revised version of the manuscript.

Comment 23:

Line 289:

"Previous research focusing on RNA dynamics and RNA binding"

I'm not sure why this is stated here. Eigenanalysis inherently reveals the strongest effects.

What is meant by a "soft mode"?

Reply

We appreciate your comment. The purpose of presenting the quoted material is to illustrate how the analysis of principal components has been applied in other regions of the spike protein, highlighting areas where the most significant variance is concentrated.

Regarding your question, a "soft mode" refers to the lowest frequency modes derived from the ANM technique. We acknowledge the importance of clarity in our terminology and will ensure that this definition is clear in the revised version of the manuscript.

“Table 3 displays the frequency values associated with the lowest frequency calculations derived from ANM, commonly referred to as soft modes.”

Comment 24:

Line 297:

"PC2 shows a high correlation (0.65) with ANM1".

In Table 3 the value is 0.52

Reply

We appreciate your understanding. We would like to clarify that the correct value is indeed 0.52. Thank you for pointing this out, and we will ensure to rectify it in the revised version of the manuscript.

Comment 25:

Figure 3:

While the authors claim significant similarity between the motions predicted by PCA and ANM, my visual impression is that the directionality is very different.

Reply

We want to express my gratitude for your careful attention to the section of our analysis in the article. Indeed, the initial impression might suggest distinct structural movements and a lack of correlation. However, we would like to clarify that the analysis conducted involves a comprehensive assessment, examining total correlations while comparing each sequence zone.

This meticulous evaluation enables us to uncover subtleties that may not be immediately evident.

Comment 27:

Figure 4:

I would attribute the correlation between the two modes to a coarse-grained correlation of large movements that hides a lot of detail.

Reply

We appreciate your thoughtful consideration of the correlation between the two modes and we concur that the observed correlation likely arises from a coarse-grained perspective, encompassing significant large-scale movements that may obscure finer details.

Comment 28:

Figure 5:

This supports my impression that only the very largest movements are to some extent represented by the analysis.

Reply

We understand that the initial observations might hint at separate structural movements lacking correlation. However, we would like to offer clarification. Our analysis encompasses a comprehensive evaluation, involving the assessment of total correlations while systematically comparing each sequence zone.

Comment 29:

Line 332:

"which is shown to be 332 consistent"

Consistent with what?

Reply

We appreciate your comment, which underscores the importance of clarity in our manuscript. In response to your suggestion, we will take measures to enhance the understanding of this section by incorporating the following paragraph into the revised version of the manuscript:

“In this study, these outcomes were employed to designate the structure for subsequent analysis, specifically the wild-type structure, aligning with the findings from the PCA and ANM investigation.”

Comment 30:

Line 335:

"structural fluctuation of Sars Cov 2, while the structures of Sars Cov 2 and Mers Cov remain immobile"

This sentence contradicts itself!

Reply

We extend our sincere apologies for the typographical error. It was an oversight on our part, and we regret any confusion it may have caused. Please rest assured that we will promptly correct this in the revised version of the manuscript.

Comment 31:

Line 360:

"excellent agreement with experimental observations"

This needs to be shown!

Reply

We are truly grateful for your meticulous attention to detail and your thoughtful question. We extend my apologies if any confusion arose, leading to a less than optimal reading experience with the manuscript.

In response to your valuable input, we have taken action by incorporating the following paragraph into the revised version of the manuscript:

“The model predicts the protein residue fluctuations and exhibits concordance between experimental measurements conducted through ANM and theoretical measurements computed by ANM.”

Comment 32:

Line 380:

Over what structures were the RMSD calculated? In many case where the authors subsequently give an RMSD, it is also not clear which structures are compared.

Reply

“An examination of the root mean square deviation (RMSD) (depicted in Figure 8 (a)) within the pre-fusion state of the HR1 structure, along with the mutations occurring in this region, unveiled a notable level of similarity among the mutant structures.”

Comment 33:

Lines 407-433:

The authors suddenly switch to the post-fusion state without giving a justification or aim. The reference to cryoEM structures is not linked to any specific results here, so why is it mentioned? What is the relevance of these results?

Reply

we are truly grateful for your meticulous attention to detail and your thoughtful question. we extend my apologies if any confusion arose, leading to a less than optimal reading experience with the manuscript.

In response to your valuable input, we have taken action by incorporating the following paragraph into the revised version of the manuscript:

Comment 34:

Line: 435:

"From the analysis of the data in the cpH- MD simulations"

What analysis? This is the first mention of MD results. How do you conclude that the pKa values are representative of the real values?

Reply

We extend my gratitude for your comment, which draws attention to the need for improved clarity in the mention of pH measurements. Your suggestion is well-received, and we are committed to enhancing the understanding of this aspect in the revised version of the article.

To address this, we will incorporate the following paragraph into the revised manuscript:

“ The titration curves presented in the Figure 11 illustrate the pKa values derived from the corresponding titration curves, as indicated in the Table 4. It is important to note that as the simulation system size decreases, the potential of the bulk water phase shifts toward a negative value. This phenomenon artificially enhances the likelihood of accepting protonation attempts and diminishes the probability of deprotonation attempts within the neMD/MC algorithm. Consequently, ionizable residues are inclined to be protonated with higher probability, resulting in an apparent upshift in the pKa values.”

Comment 35:

Figure 11:

Was the analysis done on only tripetides? That is what the legend implies.

"neutrality was maintained by adding 0.15 mol/L NaCl as buffer particles to neutralize the overall charge of the system"

NaCl is not a buffer but salt and the point to add NaCl is not charge neutralization, but rather shielding charges on the protein.

The legend should be in one paragraph, not distributed over the figure because this is confusing.

The independent axis should just be "pH".

These curves should be fit to the Henderson-Hasselbalch equation.

What does "macroscopic titration" mean?

How do these pKa values compare to typical pKa's for surface residues?

Reply

We want to express my gratitude for your thoughtful comments on Figure 11. Your insights have provided valuable guidance, and we are committed to incorporating your suggestions in the revised version of the manuscript.

We are also appreciative of the opportunity to address your questions, which will be clarified in the following paragraph that we have thoughtfully added.

“ Considering that the simulations account for numerous protonatable sites within HR1 for every variant under analysis, multiple sites will collectively contribute to a macroscopic pKa and titratable sites.”

“to ensure consistency with the probability ratio between protonated and unprotonated states as defined by the Henderson–Hasselbalch equation”

Comment 36:

Lines 488-526:

Why is this section included? It is again a switch to another topic that is not fully explained.

Reply

We are truly grateful for your meticulous attention to detail and your thoughtful question. We extend my apologies if any confusion arose, leading to a less than optimal reading experience with the manuscript.

“An exploratory study has revealed that the significance of an amphipathic secondary helix within HR1, plays a pivotal role in membrane binding. These characteristics that we were able to interpret to a certain extent from our analysis (Figure 12), thus it defines the interaction with the membrane which initiates the structural alterations in the spike protein, and we ensure that this could  facilitate efficient genome replication [75]}.”

Comment 37:

Lines 527-559:

The conclusion is too ong and not very clear. I would suggest focusing on the two key results, the classification of the structures and the pKa determinations.

Reply

We appreciate your suggestion, which highlights the importance of an efficient and clear conclusion section. Your guidance in this matter is valuable, and we are committed to improving the finality and coherence of our manuscript.

In response to your recommendation, we have incorporated the following paragraph into the conclusion section as per your suggestion:

“In this study, we performed a PCA analysis of 131 A-chain structures of the spike protein in the presence of various inhibitors. Our main objective was to explore the conformational space and inherent dynamics of the protein, with a focus on understanding ligand binding pathways for inhibitor design. We observed a strong correlation (0.72) between PCA modes and ANM modes, indicating the reliability and functional importance of low-frequency modes in adapting to different inhibitor binding.

To investigate the dynamic role of HR1 in viral processing, we used linear NMA and Non-NMA. The results showed higher RMSD ranges for linear NMA (17.02 \r{A}), indicating high inter-structure variability, whereas nonlinear NMA showed stability throughout the simulation (RMSD 4.85 \r{A}).

Analysis of the frequencies of linear and nonlinear modes revealed that the energy required for conformational changes in the nonlinear modes was significantly lower compared to the calculated linear modes. Specifically, the frequency values for the linear modes showed a value of 0.000103042 [cm$^{-1}$] for their most stable mode, whereas the nonlinear modes showed a value of 0.019 [cm$^{-1}$] for their most stable mode in HR1 structures in their prefusion form. This corresponds to a difference of 0.018896 [cm$^{-1}$] in frequencies, indicating the stability of the medians of the nonlinear modes and lower energetic costs associated with their conformational changes.

Our cpH-MD simulations revealed conformational changes in HR1 at lower pH values, indicating a shift toward post-melting structure. The pKa values determined in this study show the effects of low pH on protein structure. For example, a pKa value of 9.52 was determined for the Lys-921 residue in the D936H mutant, a value of 9.50 for the D950N mutant and a slightly higher value of 10.49 for the D936Y variant. These results are in agreement with similar trends described in other studies.

In addition, we analyzed the hydrophobic regions of the HR1 protein in the pre-fusion state to determine its hydrophobic moment in a lipid environment. The hydrophobic moment value obtained (0.565) provides information on the interactions of the protein with lipid membranes.

Further studies are needed to fully understand the functional implications of these amphipathic phases in HR1, in particular in protein-protein interactions and membrane fusion processes.”

We believe that the corrections have improved the quality of our work for which we are grateful.

Dr. Saravana Prakash Thirumuruganandham 

(https://orcid.org/0000-0003-4210-1363)

Round 2

Reviewer 3 Report

The authors addressed the detailed comments with minor modifications to the text. However, the larger justification of the analyses and simulations have not been addressed. Indeed, the authors ignored the first few paragraphs of my report. Their most extensive response was to rewrite the conclusion. However, that does not clarify why the study was done and what we should get out of it. The best of the results are consistent with the current understanding of the spike protein in virus entry, but does not add to our knowledge. I would have expected some conclusion about the nature of ligands and how they participate in the function of the spike protein, or how the pKa values impact the transformations at low pH. The important question is what experiments can one devise based on the analysis and simulations. The authors give no such indication, and therefore I have to judge this manuscript as not worthy of publication.

The English is reasonable with some minor errors, some that can be addressed with a grammar and spell checker.